**Subject Category:**
Biology (whole organism)

ecology

double-platform protocol, abundance estimation, distance sampling, perception bias, aerial surveys, SCANS-III survey

**Author for correspondence:**
C. Lambert
e-mail: charlotte.lambert@univ-lr.fr

# The effect of a multi-target protocol on cetacean detection and abundance estimation in aerial surveys

C. Lambert[1], M. Authier[1], G. Dorémus[1], A. Gilles[2], P. Hammond[3], S. Laran[1], A. Ricart[1], V. Ridoux[1,4], M. Scheidat[5], J. Spitz[1] and O. Van Canneyt[1]

[1]Observatoire PELAGIS, UMS 3462 CNRS - La Rochelle Université, 5 Allées de l'Océan, 17000 La Rochelle, France
[2]Institute for Terrestrial and Aquatic Wildlife Research, University of Veterinary Medicine Hannover Foundation, Werftstr. 6, 25761 Büsum, Germany
[3]Sea Mammal Research Unit, Scottish Oceans Institute, University of St Andrews, St Andrews KY16 8LB, UK
[4]Centre d'Études Biologiques de Chizé, UMR 7372 CNRS - La Rochelle Université, 5 Allées de l'Océan, 17000 La Rochelle, France
[5]Wageningen Marine Research, Haringkade 1, 1976CP Ijmuiden, The Netherlands

CL, 0000-0002-1128-5262

A double-platform protocol was implemented in the Bay of Biscay and English Channel during the SCANS-III survey (2016). Two observation platforms using different protocols were operating on board a single aircraft: the reference platform (Scans), targeting cetaceans, and the 'Megafauna' platform, recording all the marine fauna visible at the sea surface (jellyfish to seabirds). We tested for a potential bias in small cetacean detection and density estimation when recording all marine fauna. At a small temporal scale (30 s, roughly 1.5 km), our results provided overall similar perception probabilities for both platforms. Small cetacean perception was higher following the detection of another cetacean within the previous 30 s in both platforms. The only prior target that decreased small cetacean perception during the subsequent 30 s was seabirds, in the Megafauna platform. However, at a larger scale (study area), this small-scale perception bias had no effect on the density estimates, which were similar for the two protocols. As a result, there was no evidence of lower performance regarding small cetacean population monitoring for the multi-target protocol in our study area. Because our study area was characterized by moderate cetacean densities and small spatial overlap of cetaceans and seabirds, any extrapolation to other areas or time requires caution. Nonetheless, by permitting the

collection of cost-effective quantitative data for marine fauna, anthropogenic activities and marine litter at the sea surface, the multi-target protocol is valuable for optimizing logistical and financial resources to efficiently monitor biodiversity and study community ecology.

# 1. Introduction

Unbiased estimates of the abundance of wildlife populations are necessary for effective conservation and for the management of human activities. Several survey methodologies, associated with robust analytical methods, have been tested in the field, from complete census to quadrat, transect or strip surveys [1–4]. One of the most used non-invasive methods today is probably distance sampling, which can be land based, boat based or air based [4]. In marine systems, megafauna population estimation is mostly achieved through boat-based or aerial surveys using a dedicated protocol based on distance sampling methodology.

Ideally, abundance estimates should be derived from surveys solely dedicated to the target species (mono-target protocol), which maximizes data quality and minimizes uncertainties around distribution and abundance estimates for the targeted species [4]. The targeted species in a mono-target protocol are typically a grouping of species with similar characteristics that it is effective to survey at the same time, e.g. cetaceans or seabirds. Yet, conducting multiple single-target surveys is expensive in resources (fuel, logistics, cost, qualified personnel). In practice, multi-target surveys are increasingly being undertaken in marine systems, effectively surveying a range of taxonomic groups [5–8]. In the context of optimizing marine wildlife monitoring, multi-target surveys have the great advantage of permitting cost sharing and reducing the ecological footprint compared with mono-target surveys [9,10]. Survey data of multiple taxonomic groups collected simultaneously are also particularly relevant to understanding ecological relationships among ecosystem components, which is fundamental for the implementation of monitoring programmes [11–13].

Despite their advantages, conducting multi-target surveys comes with potential detrimental effects on the accuracy and precision of the resulting abundance estimates. These detrimental effects could arise if searching for a suite of objects with very different search images, and being potentially available all at the same time impairs the detection process. This process can be decomposed into two components: the availability of animals to detection (present at the surface or near the surface) and their perception by observers. For example, sharks and small cetaceans have different detection characteristics, sharks being harder to detect, because of their lower availability (only visible sub-surface), than small cetaceans (for which availability is higher because of their frequent surfacing). Mono-target protocols focus on one or several species with similar characteristics (the so-called target), permitting detection to be maximized through specific training of observers to the perception of this particular target. On the contrary, multi-target protocols focus on a range of taxonomic groups with varying detection characteristics.

In this case, the probability of an observer detecting an animal available at the surface could be affected when other animals with different detection characteristics are recorded. This has been suggested to be the case particularly for the detection of cetaceans while simultaneously recording seabirds. Birds flying above the water could either distract the observer from the sea surface or partially obscure the sea surface when present at very high density, thus the observer's capacity to detect an animal present at or just beneath the surface might be reduced, with a corresponding decrease in perception probability. The detection process during mono-target protocols has been extensively explored in relation to aircraft altitude, sea conditions, turbidity, animal characteristics and availability, among other factors [14–16], but the effect of detecting targets with different characteristics (e.g. seabirds, boats) compared with the detection of swimming animals (e.g. cetaceans) has not been explicitly investigated.

The detection process of marine animals and its impact on their abundance estimates has so far been investigated in three different types of experiment: tandem platforms of observation on board separate vessels or aircraft, tandem platforms of observation on board single vessels or aircraft, and a combination of visual observers with digital surveys [14,17–19]. During the large-scale SCANS-III survey in July 2016 [20], a specific double-platform protocol was implemented in the Bay of Biscay and the English Channel which consisted of two independent observer teams operating simultaneously within the same plane, similar to [14]. However, contrary to [14], each platform used a different data collection protocol. The 'Scans' platform used the same dedicated protocol as previous

SCANS surveys, and collected line-transect data on cetaceans but also recorded opportunistically the presence of non-target objects (boats, nets, marine litter, fish, turtles and jellyfish; [21,22]). The 'Megafauna' platform used a multi-target protocol to collect line-transect and strip-transect data on a full suite of target objects, from jellyfish to cetaceans and seabirds (same protocol as for the REMMOA, SAMM and ObSERVE surveys; [5–8]).

This configuration allowed exploration of the effect of surveying different targets simultaneously on the perception of small cetaceans, given they were known to be there, and on the resulting abundance estimates. Based on field experience, cetacean, fish, turtles, jellyfish and marine litter detections were hypothesized to increase the perception probability of small cetaceans because the detection of all these objects forces observers to focus their attention under the aircraft and on the sea surface. Conversely, the recording of seabirds was hypothesized to have a negative impact by distracting the observer from the sea surface for the Megafauna platform, but the potential effect of seabird detection (seen but not recorded) on small cetacean perception probability for the Scans platform was unknown.

The aim of this study was to test for bias in small cetacean (phocoenids and delphinids) detection in a multi-target protocol (the Megafauna platform) compared with a reference mono-target protocol (the Scans platform), and to test if this potential bias had an impact on abundance estimation. To do so, we first identified all unique sightings of small cetaceans (either seen by both platforms or by only one) and estimated the perception probability for each platform by implementing a capture–recapture model. We then estimated the impact of the detection of other cetaceans (either large or small cetaceans), anthropogenic objects, fish/turtles and seabirds on this perception. Second, we performed conventional distance sampling analyses on the data from each platform and compared resulting estimated effective strip widths (ESWs) and relative densities. Third, we tested the consistency of these results in cases of varying densities of observed anthropogenic objects, seabirds and fish/turtles, with a post-stratification of the effort based on observed encounter rates of these different types of sightings. Finally, we implemented a hierarchical modelling of detection functions to disentangle the confounding effects of species, observer and platform of observation on the way small cetaceans are detected.

# 2. Material and methods

## 2.1. Aerial survey and data collection

The SCANS-III survey was conducted in June–August 2016 in European Atlantic waters, from Vestfjorden, Norway, to the Strait of Gibraltar using ship-based and aerial surveys [20]. The aircraft-based double-platform protocol was conducted on transects over the Bay of Biscay and English Channel blocks, following a zig-zag layout pre-designed for the survey (figure 1). A high-wing double-engine aircraft (Britten-Norman II Islander) was equipped with bubble windows, allowing the six observers on board to operate two independent platforms of observation: the rearward 'Megafauna' platform and the forward 'Scans' platform. Communication was not possible between the two sets of observers. The six observers switched observation posts within (right and left observation window, data recording) and between platforms. All observers were trained and had previous experience with the aerial observation of marine fauna. Specific training was conducted to help the two sets of observers to easily switch between the two protocols.

The aircraft flew at a constant speed of 90 knots (167 km h$^{-1}$) and height of 600 feet (183 m). Observers recorded all environmental conditions impacting the detection of animals: sea conditions measured on the Beaufort scale (hereafter 'sea state'), turbidity, cloud cover, sky glint, glare severity, glare orientation and the 'subjective condition', which integrates all the above factors and is defined as the probability of an observer detecting a harbour porpoise. Whether swell presence impacted detection was also recorded by the Megafauna platform. All flight information and sightings were recorded using the VOR 3.2 software for the Scans platform (as used in all aerial surveys for SCANS-III), and the SAMMOA 1.0.4 software for the Megafauna platform. Each platform used an independent but similar GPS device connected to the recording software. The two protocols had a system of audio recording, ensuring that information given by observers was not lost when the number of targets detected at the same time was too large to permit real-time recording of the whole suite of information by the data recorder. In such cases, potentially unrecorded *in situ* sightings were reconstructed afterwards based on GPS and audio tracks.

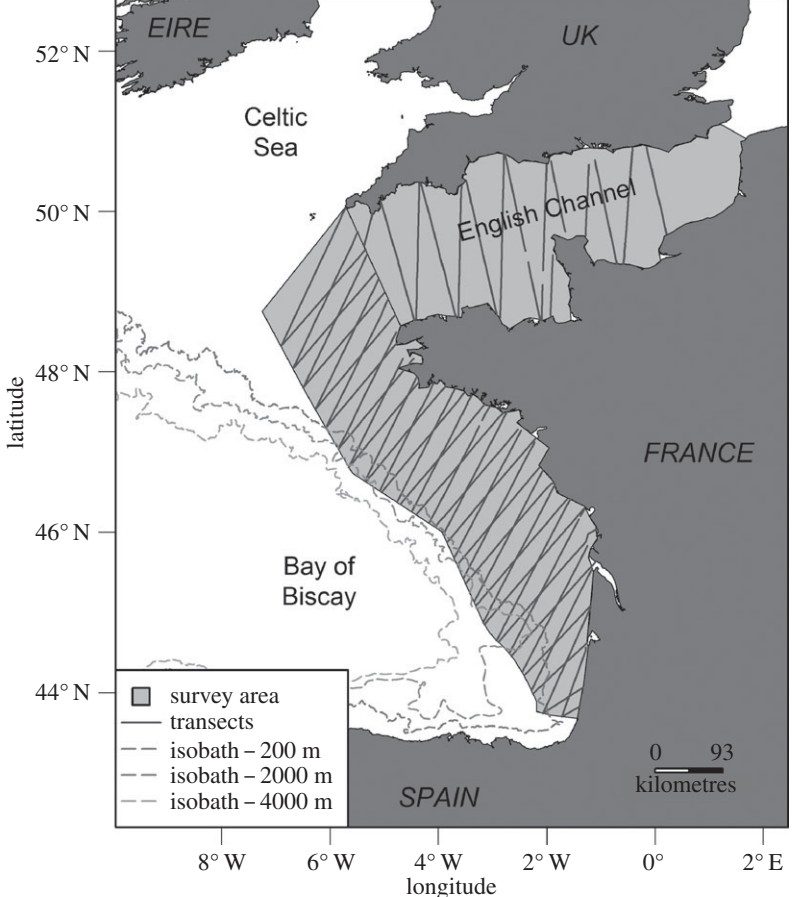

**Figure 1.** Study area, with survey area, sampled transects and isobaths.

The Scans platform followed the protocol used for SCANS aerial surveys (for details see [22,23]) and was based on line-transect (distance) sampling (figure 2*a*). Observers scanned the sea surface and sub-surface in search for cetaceans, and were encouraged to focus on the transect line. There was no maximum distance limit from the transect for the detection of cetaceans, fish, jellyfish and turtles but a maximum distance was set at 500 m for boats, nets and marine litter. The number of individuals in a sighting, detection angle (perpendicular to the track line; measured with an inclinometer), cue, behaviour, swim direction, calf presence and any animal reaction to the aircraft were recorded for each cetacean sighting. Number of items were recorded for marine litter, fishing boats, static fishing gear, turtles, large fish (sharks, tunas or bonitos, sunfish) and jellyfish within a maximum distance of 500 m from the transect.

The Megafauna platform followed line-transect methodology for cetaceans, fish and turtles, and strip-transect methodology for other targets, including seabirds, marine litter, fishing gear and boats (figure 2*b*). Observers scanned the sea surface and sub-surface as well as the air column searching for all targets. The maximum collection distance from the transect was target dependent: 200 m for seabirds, marine litter and fishing gear; 500 m for boats (fishing, commercial and leisure; strip-transect methodology). No maximum distance was defined for cetaceans, turtles, large fish (sharks, tunas or bonitos, sunfish) and jellyfish, but detection angle (as for the Scans platform) and cue were recorded. In practice, the strip-transect methodology meant that observers focused up to 500 m from the transect. Behaviour, swim direction, number of animals, presence of calves, and any reaction of the animals to the aircraft were recorded. Seabird activity (resting or flying) was also recorded. When possible, the (st)age of sighted individuals was assessed for species with distinctive (st)age-dependent patterns, such as gannets and black-legged kittiwakes.

## 2.2. Data processing

Total distance flown on effort was 10 425 km for the Megafauna platform and 10 437 km for the Scans platform. This difference was due to time lags at the beginning and end of the survey effort, because

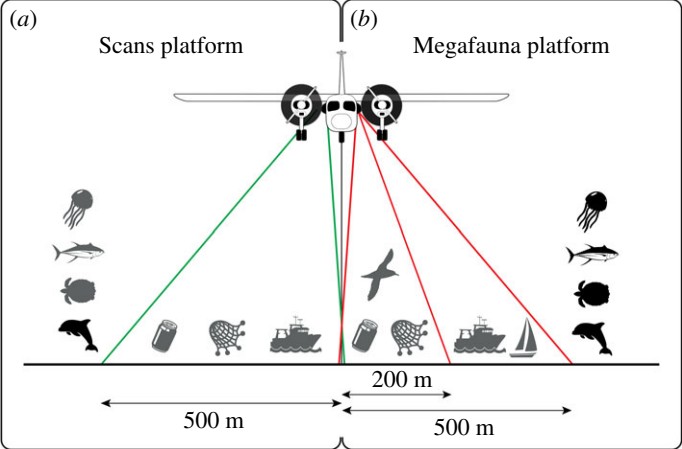

**Figure 2.** Survey protocol for the two observation platforms. (*a*) Scans platform: observers scanned the sea surface and sub-surface without a distance limit but focusing on the transect line in search of cetaceans while also recording any fish, turtles, fishing boats, fishing gear or marine litter sightings. The three latter items were recorded only within a 500 m limit. (*b*) Megafauna platform: observers scanned the sea surface and sub-surface as well as the air column below the aircraft and recorded all sighted items; seabirds, marine litter and fishing gear were recorded within a band of 200 m, boats within a band of 500 m, and cetaceans, fish, turtles and jellyfish were recorded without a distance limit but with a detection angle (in practice the strip-transect methodology implies that observers focus on the 500 m from the transect). The species available for distance sampling analysis (i.e. for which detection angles were recorded) are represented by symbols in black.

the platforms were independent. The majority of the survey effort (84% and 74% for Megafauna and Scans platforms, respectively) was conducted in good observation conditions, i.e. with sea state less than 4 and subjective conditions from moderate to excellent.

The small cetacean perception probability was estimated based on sighting information only, pooling all small cetacean sightings recorded on effort: harbour porpoise (*Phocoena phocoena*), common dolphin (*Delphinus delphis*), striped dolphin (*Stenella coeruleoalba*), bottlenose dolphin (*Tursiops truncatus*) and unidentified delphinids. Density estimates were derived from transects of effort, subdivided into legs of homogeneous observation conditions. For each leg, the number of sighted individuals was summarized for the following groups of species: marine mammals (large and small cetaceans), harbour porpoise, common and striped dolphins, bottlenose dolphin. Table 1 summarizes the species considered for each analysis.

## 2.3. Descriptive statistics

Sightings were classified into four main categories of interest for this study: anthropogenic objects (marine litter, fishing gear), marine mammals (mostly cetaceans), other marine fauna (not marine mammals) and seabirds. Encounter rates were computed for each platform and category as the ratio of the total number of sightings over the sum of effort within the study area. Legs of effort were further subdivided into 10-km-long segments to homogenize the length of sampling unit and to allow for a fine-scaled examination of differences between the two platforms. Segments were paired by temporal matching between the Megafauna and Scans platforms, and the difference in encounter rates between the two platforms per category was computed for each segment.

## 2.4. Small cetacean perception

### 2.4.1. Identification of duplicates

To quantify the potential effect of the detection of other targets on the platform-specific small cetacean perception probability, we first identified the set of unique small cetacean sightings.

Identifying unique sightings required determining which sightings were seen by both platforms (hereafter 'duplicates') and which were seen by one platform only. We implemented an automated routine in R-3.4.3 [24] based on an objective decision tree (figure 3). This decision tree provided similar results irrespective of choice of the focal platform. A sighting from platform 2 was tested for

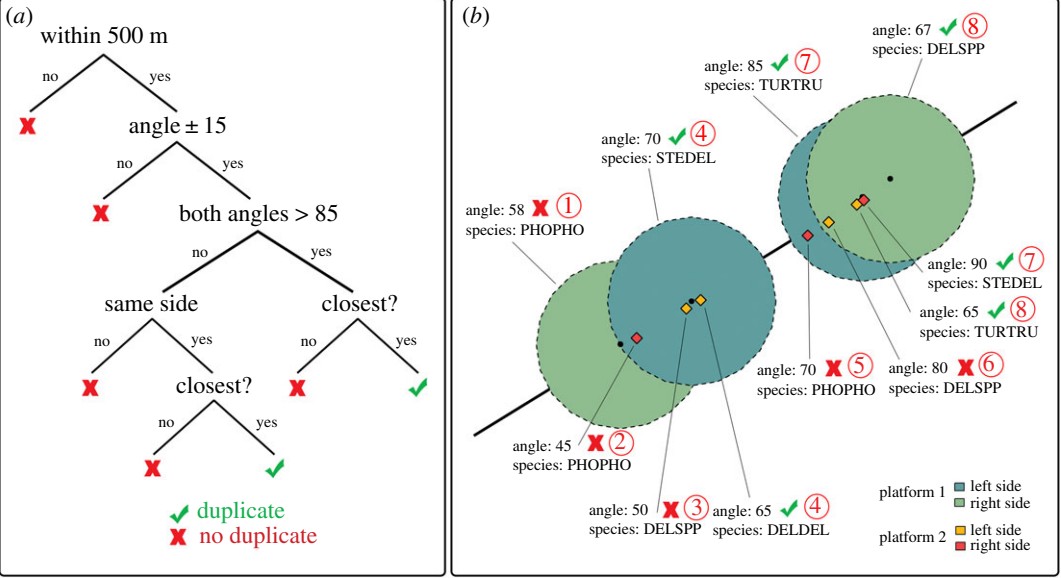

**Figure 3.** Duplicate identification. (*a*) Decision tree. (*b*) Example. The black line is the surveyed transect, and the black dots and diamonds are the platform 1 and 2 sightings' recording positions on the transect, respectively. Circles are 500 m searching areas around focal sightings from platform 1 (left side in blue, right side in green), and diamonds are platform 2 sightings (left side in yellow, right side in red). Unique sighting numbers are shown in red (duplicate numbers indicate duplicate sightings), and identified duplicate sightings are shown with green tick marks.

**Table 1.** Species groups and sampling unit used for each analysis. '&' symbols indicate that groups were considered together, '|' symbols that species were considered separately in the analysis.

| analysis type | groups considered in analyses | sampling unit |
| --- | --- | --- |
| encounter rates | anthropogenic objects (marine litter, fishing gear) \| marine mammals \| other marine fauna (except marine mammals) \| seabirds | 10 km segments |
| small cetacean perception | harbour porpoise & common dolphin & striped dolphin & bottlenose dolphin & unidentified delphinid/phocoenid | sightings |
| conventional distance sampling | harbour porpoise \| common dolphin & striped dolphin \| bottlenose dolphin \| marine mammals | legs of homogeneous detection conditions |
| post-stratification | harbour porpoise & common dolphin & striped dolphin & bottlenose dolphin & unidentified delphinid/phocoenid | legs of homogeneous detection conditions |
| observer effect | all cetaceans | legs of homogeneous detection conditions |

duplication only if it was recorded within an omnidirectional 500 m buffer around the focal sighting from platform 1. This threshold was chosen to account for uncertainty in using independent GPS for each of the two platforms (500 m is the usual uncertainty associated with GPS localization) and uncertainty inherent to the recording process (the small delay between detection by the observer and the actual recording by the data recorder, which was not greater than 100 m, on average).

If the detection angle of a candidate sighting was not included within an interval of 15° around the focal sighting detection angle (detection angles are angles of declination recorded perpendicular to the track line), the candidate sighting was not classified as a duplicate of the focal sighting. Otherwise, the candidate sighting was further assessed: if the detection angle was greater than 85° (sightings under the plane), the candidate sighting was considered a duplicate. If the detection angle was less than 85° (further from the line, up to the horizon), the candidate sighting was considered a duplicate if it was recorded on the same side of the aircraft as the focal sighting. If several sightings were concurrently identified as potential duplicates of the same focal sighting, only the closest one was classified as a duplicate.

The 15° criterion was necessary because of unavoidable uncertainty in measuring the declination angle, which could be due to observers spotting different animals in the sighted group and to uncertainty in assessing exactly when the sighted individuals crossed the line perpendicular to the transect.

This decision tree ignored the observer side for sightings made directly under the aircraft (i.e. angle larger than 85°), because such sightings are detectable from both sides of the aircraft. Species identity was not considered because of the underlying variation in observation conditions. For example, two observers might not record the same animal with the same taxonomic precision (e.g. common dolphin versus unidentified small delphinid) depending on the detection cues they could pick.

### 2.4.2. Perception probability estimation

For cetaceans, detection probability $p$ is a function of the animals' availability to detection (present at the surface or sub-surface) and their perception by observers. Because unique sightings of small cetaceans were seen by at least one platform, animals were both present and available for detection. Availability was assumed to be the same between platforms, justified by the bubble windows being of similar width and shape and thus providing equivalent fields of view between the platforms. The time lag between platforms was very short, allowing the field of view to be assumed to be the same during the time the aircraft flew over the animals. As a consequence, the detection probability $p$ estimated here equals the perception probability.

A capture–recapture model applied to tandem observation platforms was used to estimate the detection probability of small cetaceans. More specifically, occupancy models were fitted [25], considering the two platforms as independent sampling sessions and unique small cetacean sightings as equivalent to sampling sites. This setting is conceptually equivalent to a two-visit survey where all sites are occupied (occurrence $\psi = 1$), and allowed us to focus on the platform-specific detection probabilities $p$ of small cetaceans.

Small cetacean perception probability $p$ was modelled as a linear function of variables with a logit link function. The general form of the model was logit $(p_{ij}) = \alpha + \beta_x x_{ij}$, where $p_{ij}$ is the perception probability for cetacean sighting $y_{ij}$ on platform $j$, $x_{ij}$ represents whether or not (0, 1) a sighting of another object of class $x$ was detected shortly before cetacean sighting $i$ by platform $j$ on a leg of effort and $\beta_x$ is the effect of variable $x$ (see below).

We started with a null model to estimate the overall perception probability of small cetaceans by platform, irrespective of the detection of other targets. Then, we tested the effect of detecting other targets on the detection of small cetaceans $p$ by including covariates that were indicator variables of whether other targets were sighted during the 30 s prior to each unique small cetacean sighting. The 30 s threshold was chosen based on observer experience, from which the detection of a sighting was believed to have no effect on subsequent detection after this lag, and must be adjusted to particular situations when transferring the method to other study areas and species. The targets were divided into six categories: cetaceans (not restricted to small cetaceans); other marine fauna (excluding cetaceans: fish, turtles, jellyfish); anthropogenic objects (marine litter, fishing gear); flying seabirds; resting seabirds (standing on water); and large groups of seabirds (greater than 10 individuals, either flying or resting).

For each unique sighting of small cetaceans (i.e. seen by at least one platform), the detection of cetaceans, other marine fauna, anthropogenic objects and seabirds during the previous 30 s was assessed for each platform based on its recorded sightings (figure 4). The presence of seabirds on the transect for the Scans platform was assessed using the Megafauna platform seabird sightings as a proxy. The strip-transect protocol used to record seabirds in the Megafauna platform hypothesized perfect detection of seabirds over the strip, hence seabird sightings from the Megafauna platform can be used to infer seabird presence prior to cetacean sightings on the Scans platform.

Occupancy models were fitted with the package `unmarked` [26] in R-3.4.3 [24]. To test for a potential platform-dependent effect of target detection, we fitted a set of models with only simple effects of covariates and a set of models with interaction terms with the platform:

$$\text{model 1}: p \sim \text{Platform} + \text{Cetaceans} + \text{Other Marine Fauna} + \text{Anthropogenic Objects}$$
$$+ \text{Flying Seabirds} + \text{Resting Seabirds} + \text{Seabirds In Groups}$$
$$\text{model 2}: p \sim \text{Platform} + \text{Cetaceans} + \text{Other Marine Fauna} + \text{Anthropogenic Objects}$$
$$+ \text{Flying Seabirds} + \text{Resting Seabirds} + \text{Seabirds In Groups}$$
$$+ \text{Platform}:\text{Cetaceans} + \text{Platform}:\text{Other Marine Fauna}$$
$$+ \text{Platform}:\text{Anthropogenic Objects} + \text{Platform}:\text{Flying Seabirds}$$
$$+ \text{Platform}:\text{Resting Seabirds} + \text{Platform}:\text{Seabirds In Groups}.$$

**Figure 4.** An example of target detection assessment for a unique sighting of a small cetacean. Small cetacean unique sighting (shown with the circled arrowhead) has been detected only by platform 1 (yes). Seabird, anthropogenic objects, other marine fauna and cetacean detections were assessed within the 30 s preceding the unique sighting. The detection of targets informs the table used for $p$ estimation: 0, no target detected; 1, at least one target detected.

We predicted, based on field experience, a positive impact of cetacean, other marine fauna and anthropogenic object detections on small cetacean perception probability because these tend to focus the observer's attention under the aircraft and on the sea surface. However, for the Megafauna platform we expected a reduced positive effect on small cetacean perception probability for other marine fauna since this protocol necessitates recording a detection angle for these targets (i.e. distraction from the sea surface), while the Scans protocol does not. For the Megafauna platform, flying seabirds and flocks of seabirds could have a negative impact by distracting the observer from the sea surface, but resting seabirds should have an effect similar to that of cetacean, marine fauna and anthropogenic objects. The potential effect of seabird detection (seen but not recorded) on small cetacean perception probability for the Scans platform is unknown by design.

### 2.4.3. Sensitivity analysis

We tested whether the above results were sensitive to the set of sightings identified as duplicates by changing the threshold values of the decision tree. We modified the second threshold, the detection angle, to test two extreme values: sightings were tested for duplicates if the difference between their detection angle was less than 5° (a more restrictive tree) or 25° (a less restrictive tree). The perception probability was then estimated with the two sets of identified unique sightings as described above.

## 2.5. Small cetacean density estimation

### 2.5.1. Conventional distance sampling

Conventional distance sampling (CDS) analyses were conducted for all marine mammals (large and small cetaceans), harbour porpoises, common/striped dolphins and bottlenose dolphins using package `Distance` [27] in R-3.4.3 [24].

For marine mammals, harbour porpoises and common/striped dolphins, only sightings made under good observation conditions (sea state less than 4 and moderate to excellent subjective conditions) were included in the analysis; all sightings were used for bottlenose dolphin. Frequency histograms of perpendicular distance were inspected visually and truncated to remove sightings made far from the transect line (outliers). We truncated sightings at perpendicular distances as follows: marine mammals—680 m for the Megafauna platform, 530 m for the Scans platform; common/striped dolphins—400 m for both platforms; harbour porpoises—280 m for the Megafauna platform, 290 m for the Scans platform; bottlenose dolphins—340 m for the Megafauna platform, 410 m for the Scans platform. We tested half-normal and hazard rate keys for detection functions [4], and selected the best models using the Akaike information criterion.

ESWs and densities over the whole study area were estimated and compared between platforms for each group.

### 2.5.2. Post-stratification

The next step was to explore potential differences in estimated ESWs and the density of pooled small cetaceans (harbour porpoises, common, striped and bottlenose dolphins, and unidentified delphinids)

**Table 2.** Threshold encounter rates for effort post-stratification by platform, for anthropogenic objects, seabirds and other marine fauna. Megafauna values were used to determine classification; Scans values are encounter rates observed in the corresponding classes for this platform.

| target type | class | Megafauna | Scans |
|---|---|---|---|
| anthropogenic objects | 1 | 0.00 | [0.00; 1.16] |
| | 2 | [0.07; 0.17] | [0.00; 1.44] |
| | 3 | [0.17; 0.30] | [0.00; 4.19] |
| | 4 | [0.30; 0.50] | [0.00; 1.80] |
| | 5 | ≥0.50 | [0.00; 2.50] |
| seabirds | 1 | 0.000 | — |
| | 2 | [0.07; 0.12] | — |
| | 3 | [0.12; 0.28] | — |
| | 4 | [0.28; 0.52] | — |
| | 5 | ≥0.528 | — |
| other marine fauna | 1 | 0.00 | [0.00; 0.41] |
| | 2 | ≥0.07 | [0.00; 0.51] |

between the two platforms as a function of the relative density of different groups of detected target types. To achieve this, we post-stratified the effort based on the encounter rates of anthropogenic objects (marine litter, fishing gear, boats), seabirds (pooling flying and resting birds) and other marine fauna (excluding cetaceans: fish, turtles, jellyfish).

Five classes were created for each target type. Class **1** was based on zero encounter rates, classes 2–5 were based on non-zero encounter rate quartiles as follows—class **2** minimum–2nd quartile; class **3**: 2nd quartile–median; class **4**: median–3rd quartile; class **5**: ≥3rd quartile. Only two classes were constructed for other marine fauna due to lower encounter rates—class **1**: zero encounter rates, class **2**: non-zero encounter rates. Classes were created based on encounter rates from the Megafauna platform (table 2); classes for the Scans platform were then created by temporal matching of segments between the two platforms (hence segments within each class were spatially consistent between platforms).

Encounter rates, ESWs and densities for each class and platform were estimated by CDS as described above (§2.5.1), for pooled small cetacean species (harbour porpoises, common, striped and bottlenose dolphins, and unidentified delphinids). Only observations made in good conditions were considered (sea state less than 4 and moderate to excellent subjective conditions).

### 2.5.3. Observer effect exploration

Our aim here was to quantify the platform effects on detection functions, while accounting for observer and species, through an analysis of variance of ESWs of each combination of platform–observer–species.

A hierarchical model with cross-classified random effects of observers and species for each platform was built assuming a half-normal key for the detection function. Random effects allow the leverage of partial pooling (a.k.a. borrowing strength) by shrinking each combination of platform–observer–species detection function towards a common mean detection [28]. If, for a given combination of parameters, there are few sightings, the estimated detection function will be very close to the common detection function, whereas if there are enough data the estimated detection function can deviate from this common function. In other words, random effects allowed us to stabilize the estimated detection function even in cases of sparse data for some combinations of platform–observer–species.

For sighting $i$ of species $k$ from platform $l$ by observer $j$, let $d_{ijkl}$ denote the perpendicular distance. The detection probability of species $k$ by observer $j$ on platform $l$ is:

$$p_{ijkl} = g_{jkl}(d_{ijkl})$$

$$= e^{-((d_{ijkl})/(\sqrt{2}\sigma_{jkl}))^2}$$

$$\log(\sigma_{jkl}) = \delta_l + \eta_{jl} + \gamma_{kl},$$

where $\delta_l$ is the intercept for platform $l$, $\eta_{jl}$ is the effect of observer $j$ on platform $l$ and $\gamma_{kl}$ is the species-specific effect on detection from platform $l$. The subscript $i$ indexes a sighting with its associated perpendicular distance $d_{ijkl}$ (in kilometres). $\eta_{jl}$ and $\gamma_{kl}$ were modelled with bivariate Normal random effects, specified with a Cholesky decomposition and using priors for the Cholesky factors from [29]. Using a bivariate effect makes allowance for a correlation between platforms as the same observers rotated across platforms, and the same species could be detected on each platform. Normal distributions were parameterized in terms of location (mean $\mu$) and scale (standard deviation $\sigma$) parameters: $\mathcal{N}(\mu, \sigma)$. We used half Student-$t$ distributions with three degrees of freedom and scale set to 1.5 as priors for the dispersion parameters, and standard normal priors for all other parameters (see the code in electronic supplementary material, File A). The hierarchical model was written in Jags v.4.0.0 and fitted with R-3.4.3 [24] using package `rjags` [30]. Four chains were run with a warm-up of 10 000 iterations, followed by another 10 000 iterations (with a thinning factor of 10). Parameter convergence was assessed with Gelman–Rubin $\hat{R}$ statistics. Posterior inferences were based on the pooled sample of 4000 values (1000 per chain). The ESW for each combination of platform–observer–species was computed as ($x$ is the perpendicular distance, $w_k$ is the truncation distance for species $k$):

$$\text{ESW}_{jkl} = \int_0^{w_k} g_{jkl}(x)\,dx = \int_0^{w} \left[ e^{-(x/(\sqrt{2}e^{\delta_l+\eta_{jl}+\gamma_{kl}}))^2} \right] dx.$$

# 3. Results

## 3.1. Descriptive statistics

Anthropogenic objects were the most sighted items from both platforms, with encounter rates of 0.45 and 0.37 sightings per km for the Scans and Megafauna platforms, respectively (figure 5$a$). Marine mammals and other marine fauna had similar encounter rates over both platforms (0.06 and 0.07 sightings per km for each target type for Scans and Megafauna platforms, respectively). The seabird encounter rate for the Megafauna platform was intermediate at 0.24 sightings per km (figure 5$a$). Species-specific encounter rates and the spatial distribution of sightings for each group are given in electronic supplementary material, File B. Small cetaceans (harbour porpoises, common and striped (pooled as 'small-sized delphinids'), bottlenose and unidentified dolphins) represented 95% and 95% of marine mammal sightings for the Scans and Megafauna platforms, respectively.

Differences among groups and across the two platforms in segment-based encounter rates were small. The mean difference in encounter rates between platforms was null for marine mammals and other marine fauna (figure 5$b$). The mean difference was slightly negative for anthropogenic objects (i.e. a higher encounter rate for the Scans platform), but was not statistically different from zero (figure 5$b$).

## 3.2. Perception probability estimation

A set of 1047 unique small cetacean sightings was identified. Among these, 37% were seen by both platforms, 29% only by the Scans platform and 34% only by the Megafauna. The estimated overall perception probability (from the null model) was thus slightly higher for the Megafauna platform (0.76) than for the Scans platform (0.65).

The difference in small cetacean perception probability between the two platforms in the absence of previous target detection was statistically significant ($\hat{p} = 0.40$ for the Scans platform, $\hat{p} = 0.56$ for the Megafauna platform; figure 6). The effects of previous targets were significant and also platform dependent, as models with interaction terms were the best ones.

Cetaceans and anthropogenic objects had a positive impact on small cetacean perception probability (figure 6$a$). A previous detection of cetaceans increased the perception probability by 0.54 (from 0.40 to 0.94) on the Scans platform and by 0.41 (from 0.56 to 0.97) on the Megafauna platform. A previous detection of any anthropogenic object increased the perception probability by 0.30 (from 0.39 to 0.69) and by 0.20 (from 0.56 to 0.76) on the Scans and Megafauna platforms, respectively. The effect of other marine fauna detections was positive for the Megafauna platform ($p = 0.78$), but no effect was detected for the Scans platform (figure 6$a$).

The effects of seabirds were more contrasted (figure 6$b$). Detection of flying seabirds decreased the perception probability (to 0.24) while detection of resting seabirds and seabirds in groups increased it on the Megafauna platform (to 0.76 and to 0.82 for resting seabirds and seabirds in groups, respectively).

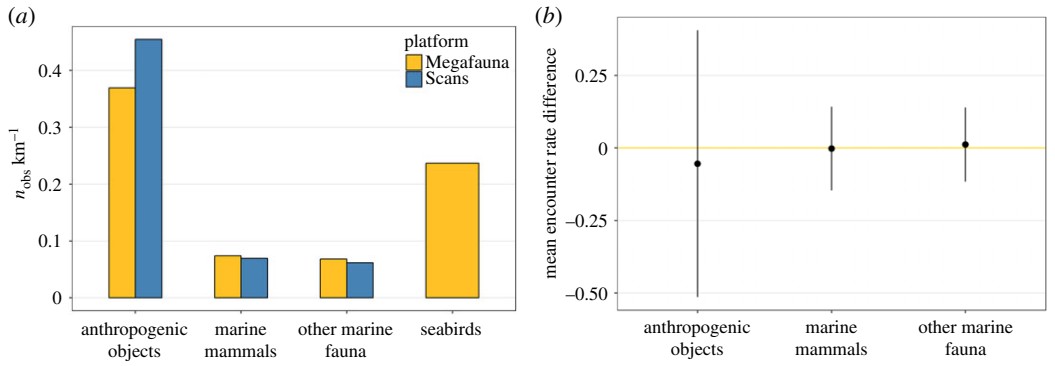

**Figure 5.** (*a*) Encounter rates for anthropogenic objects, marine mammals, other marine fauna for each platform and seabirds for the Megafauna platform over the study area. (*b*) Mean (and standard deviation) difference in encounter rate for paired Megafauna–Scans segments for anthropogenic objects, marine mammals and other marine fauna. Negative values correspond to higher encounter rates for the Scans platform.

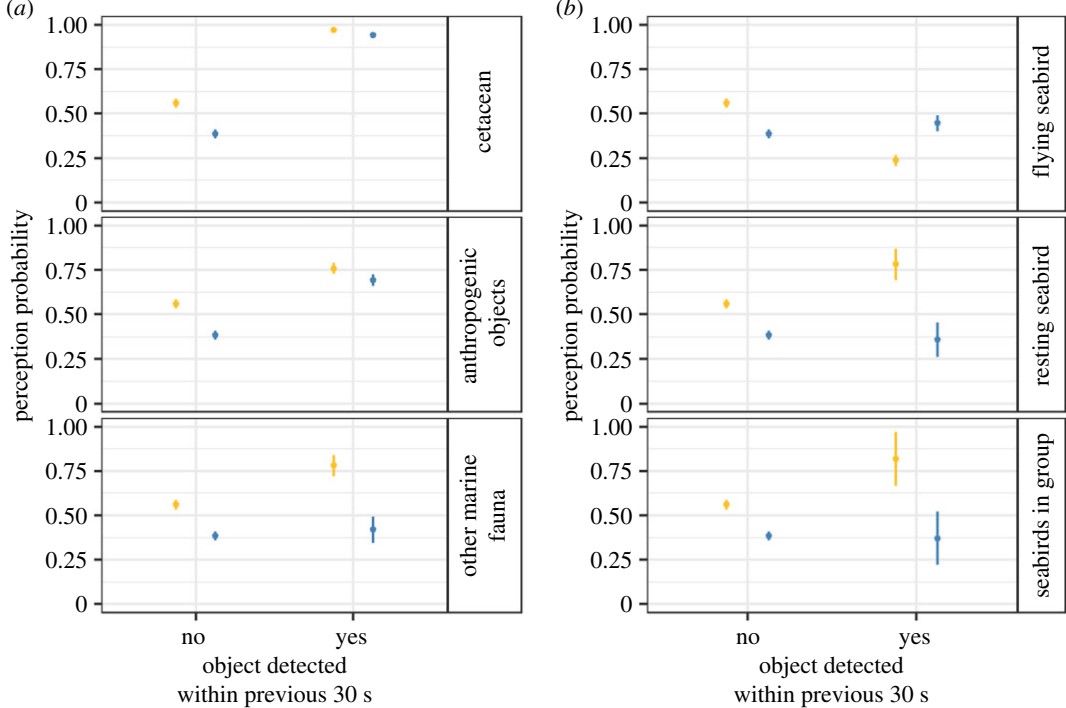

**Figure 6.** Small cetacean perception probability (estimated mean and standard error) by platform given the absence (no) or presence (yes) of detection of other items during the 30 s before the small cetacean sighting occurrences. (*a*) Effects of cetacean, other marine fauna and anthropogenic objects. (*b*) Effects of flying seabirds, resting seabirds and seabirds in groups. Megafauna platform values are in gold, Scans platform values in blue. Vertical bars represent the 95% confidence interval.

The latter two effects were imprecisely estimated (note the large confidence intervals in figure 6*b*). The effects of seabirds were different on the Scans platform (figure 6*b*), with a slightly higher (but not statistically different) perception probability when flying seabirds were present (from 0.40 to 0.45). The presence of resting seabirds or seabirds in groups in contrast had no effect. Seabirds (either flying, resting or in groups) had no effect on the perception of small cetaceans by the Scans platform.

The sensitivity analysis for the perception probability estimation indicated good confidence in the estimation of the overall perception probability for each platform (from the null model; electronic supplementary material, File C). With both tested thresholds, the estimated perception probability was slightly higher for the Megafauna platform ($p = 0.61$ with the 5° threshold; $p = 0.73$ with the 25° threshold) than for the Scans platform ($p = 0.57$ with the 5° threshold; $p = 0.68$ with the 25° threshold). Both analyses confirmed the positive impact of previous cetacean detection on the perception

probability. However, the other effects presented above were all reduced, and the negative impact of flying seabird detection was attenuated.

## 3.3. Small cetacean density estimate

### 3.3.1. Conventional distance sampling

The half-normal key was selected as the best fit for detection functions of all taxonomic groups. CDS-based ESW estimation indicated that there was no platform effect on ESWs for any of the taxonomic groups (overlapping confidence intervals, figure 7; see electronic supplementary material, File D for associated detection functions (figure 1) and detailed CDS results (table 1). For marine mammals, ESW estimates were similar for the Megafauna platform (213 m) and the Scans platform (225 m; figure 7). Harbour porpoises and common/striped dolphins had similar ESWs, while bottlenose dolphins had a slightly larger one, also with a larger uncertainty. ESWs were similar between the two platforms for the four taxonomic groups.

CDS-based estimated densities over the study area for the four taxonomic groups were similar between platforms. Common/striped dolphins had a much greater density than harbour porpoises and bottlenose dolphins and represented about 80% of the marine mammal estimated density for both platforms.

### 3.3.2. Post-stratification

A strong overlap in encounter rate, ESW and density estimates between platforms occurred for all classes of the three target types (figure 8). This overlap indicates a high level of similarity in estimates between platforms whatever the density of targets. Small cetacean species composition of sightings did not differ across classes, for any of the three target types (electronic supplementary material, File D, figure 2).

### 3.3.3. Observer effect exploration

Hierarchical modelling enabled us to partition the variance in detection into platform, observer and species components (figure 9; detailed detection functions are shown in electronic supplementary material, File D; figure 3 for the Megafauna platform, figure 4 for the Scans platform). For both platforms, the greatest variance was associated with species, and the smallest with observers. Hence, the detection functions of different species were very different, but were not very different between platforms or between observers for a given species.

# 4. Discussion

The double-platform protocol was implemented during the SCANS-III survey in 2016 to test for a potential bias in small cetacean estimation when operating the megafauna protocol used during several marine aerial surveys (e.g. the REMMOA, SAMM and ObSERVE surveys; [5–8]). This protocol could result in inaccurate or imprecise population estimates of small cetaceans if recording other taxa with different detection characteristics introduced bias in small cetacean detectability.

The double-platform protocol was previously tested during two pilot studies in 2012 and 2014 (SAMM and SAMM-ME), but the amount of effort achieved resulted in too few sightings to allow sufficient statistical power to reliably identify any potential bias in small cetacean detection due to the detection of other animals (see electronic supplementary material, File E; [6,31]). The amount of effort and sightings with the double-platform protocol during SCANS-III (greater than 10 400 km and greater than 700 cetacean sightings) was far greater than previous surveys (greater than 2000 km and less than 200 cetacean sightings), allowing us to identify clear patterns in small cetacean detections within and between the two tested protocols.

The detection probability estimated here equals the perception probability of small cetaceans because availability bias was assumed to be constant across platforms. According to the literature, small cetacean perception probability is survey dependent, ranging from 47% to 96% [15,16,32]. The perception probabilities estimated for the SCANS-III survey were within this range, and varied between the two platforms. Our results highlighted a slightly higher perception probability of small cetaceans on the Megafauna platform than on the Scans platform, irrespective of any detection of other targets. We can only speculate about the reasons for this difference. It might originate in the observer focusing better

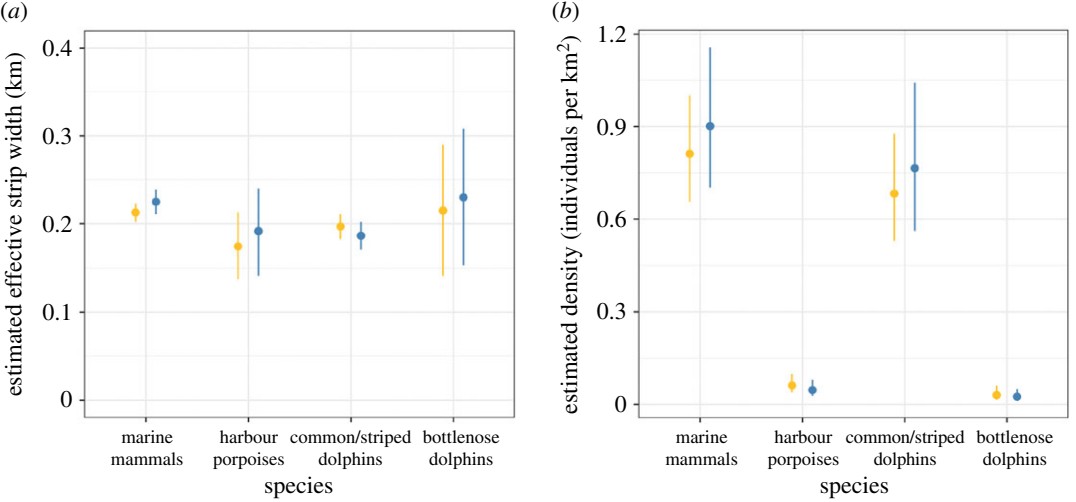

**Figure 7.** Effective strip widths (*a*) and relative density (*b*) estimated through CDS for all marine mammals, harbour porpoises, common and striped dolphins and bottlenose dolphins. Megafauna platform values are in gold, Scans platform values in blue. Vertical bars represent the 95% confidence interval.

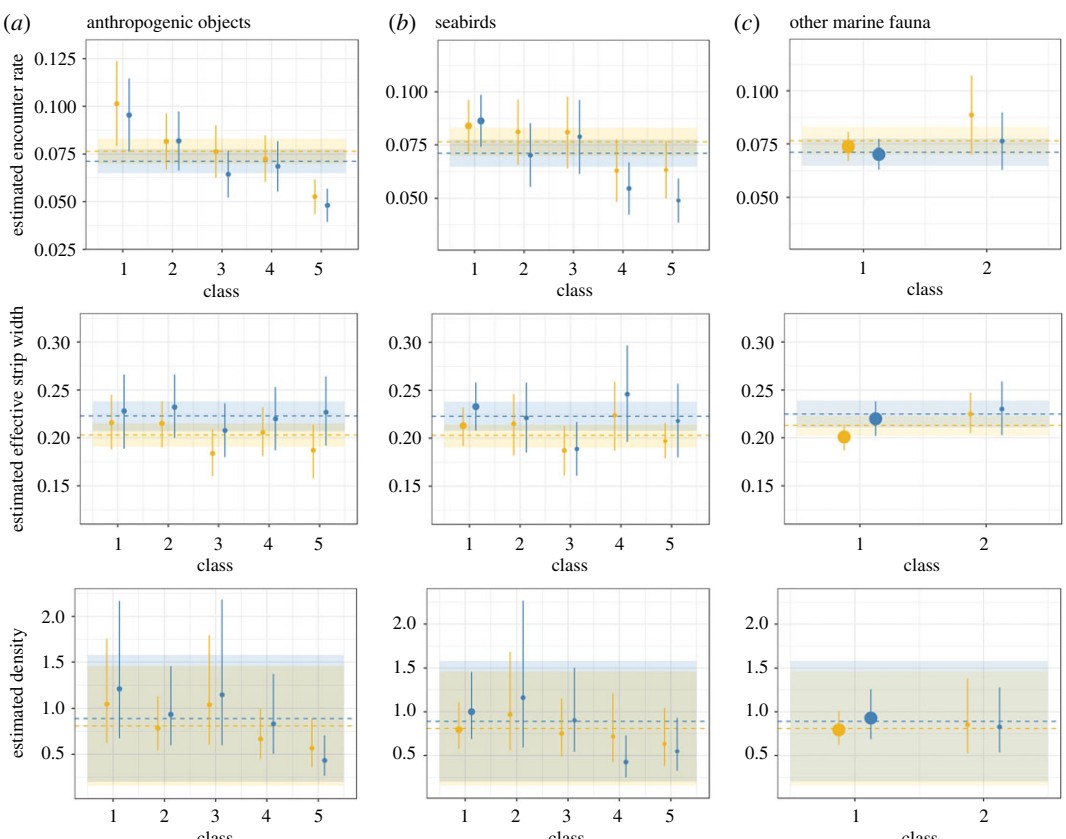

**Figure 8.** Mean small cetacean encounter rate (number of groups per km), ESW (km) and density (number of individuals per km$^2$) estimated by CDS by post-stratified class of anthropogenic objects (*a*), seabirds (*b*) and other marine fauna (*c*). Megafauna platform values are in gold, Scans platform values in blue. Dotted lines and ribbons indicate mean value and associated uncertainty estimated with the full dataset for Megafauna (gold) and Scans (blue) platforms. Point size is proportional to the total effort in each class. Vertical bars represent the 95% confidence interval.

on the transect while operating the Megafauna protocol, as shown by an overall higher detection rate. Observers' feedback suggests that the different positions of the observation platforms within the plane might partially explain this difference. The rearward platform (Megafauna) took advantage of the airplane wing located directly above the bubble-windows, protecting the observer from direct

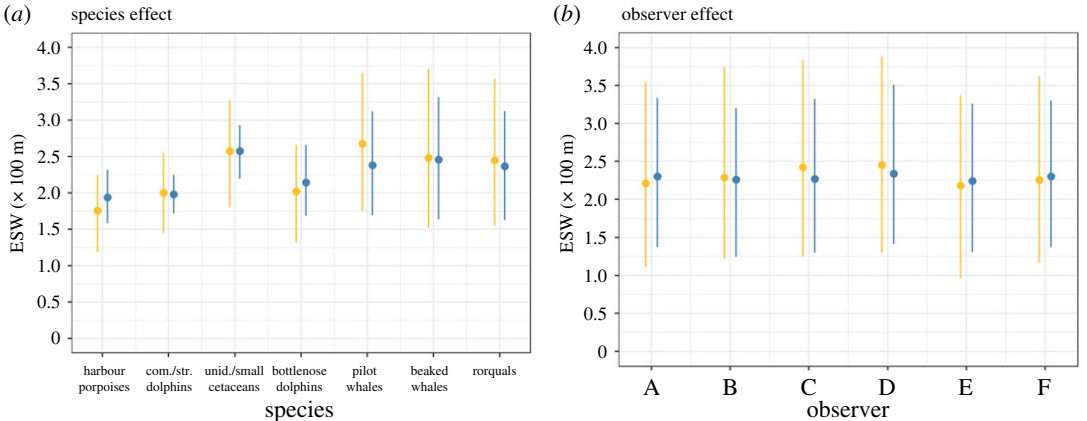

**Figure 9.** Summary of species (*a*) and observer (*b*) effects on ESWs and associated coefficients of variation estimated for each platform (Megafauna platform in gold, Scans platform in blue). For species effect, ESWs were estimated for each species but pooling all observers; for observer effects, ESWs were estimated for each observer but pooling all species. Vertical bars represent the 95% confidence interval.

sunlight. Observation conditions were considered more comfortable while operating this platform in contrast to the forward platform (Scans), which was exposed to direct glare. Unfortunately, we were not able to directly test this difference. Operating a similar double-platform experiment as in the present study but shifting the position of the two platforms inside the plane would allow us to test for the effect of direct glare on perception.

Perception of small cetaceans has been demonstrated to be affected by environmental observation conditions, and to be different among species [14,32,33]. Here, we assessed, for the first time, the effects of the presence of other targets on small cetacean perception. Such effects were strongly dependent on the type of target considered. Cetacean presence induced a higher perception probability for both platforms. This effect was clearly the strongest identified effect and was robust to variations in the assumptions made to identify duplicates. It could be due to a real impact of cetaceans on the detection process by focusing the observer on the sea surface. This impact of cetaceans on the detection process might be driven by a non-random distribution of small cetaceans, known to exhibit aggregative behaviours, whereby the detection of one small cetacean group will tend to increase the probability of detecting another one in a short period of time, thereby keeping the observer focused on the sea surface. How influential this is would depend on the spatial scale of aggregation of these species, about which little is known.

In European shelf seas, it has been suggested that the presence of birds could divert the attention of observers away from cetaceans or even be present in densities such that they could partially obscure the sea surface and lower the perception of cetaceans, so we expected a negative effect of seabird detection on both platforms in our study area. Flying seabirds had a negative effect on small cetacean perception only on the platform recording them (the Megafauna platform). As a result, in our study area, this negative effect might originate in a switch of focus from the sea surface to the air column but not from masking the sea surface, in which case the negative effect would occur in both platforms. Indeed, areas of highest seabird densities generally did not overlap with higher density areas for cetaceans (seabirds were mostly inshore, whereas cetaceans were mostly offshore; [34]). Hence, this small-scale perception bias might have a negligible impact on high-density cetacean areas. In addition, the negative impact of seabirds on cetacean perception for the Megafauna platform was probably counterbalanced by the overall higher perception probability for this platform, but would affect spatially explicit models and inference into community ecology.

We expected to find that detection of other marine fauna (large fish, turtles, jellyfish) would have a positive impact on small cetacean perception probabilities for both platforms. However, this was true only for the Megafauna platform; for the Scans platform, we found no effect.

Unlike the overall platform-dependent perception probability and cetacean detection effect, the effects of seabirds, anthropogenic objects and other marine fauna detection were reduced when varying the assumptions made to define whether a sighting was a duplicate or not. As a consequence, our results strongly indicated that the Megafauna platform does not perform poorly compared with the Scans platform regarding small cetacean perception, and that, although cetacean detection increases the perception probability of small cetaceans, some caution has to be exercised regarding the beneficial

effect of seabirds, anthropogenic objects and other marine fauna detection. The sample size used to infer these perception probabilities was fairly large (1047 unique sightings) and sufficient to clearly identify large patterns (overall perception probability), but possibly a bit too low to identify robust patterns when stratifying the perception process into six categories. Further refinement with an extended dataset would be advisable to confirm or inform the negative seabird effect identified here.

Analyses to estimate detection functions gave similar results for the four taxonomic groups, in terms of both the detection functions themselves (there was no variation in the way observers detected cetaceans between platforms) and the estimated population densities. When post-stratifying, we found that seabird and other target densities impact neither the way small cetaceans are observed (ESWs) nor the final estimated density. Thus, the negative impact at 30 s identified above was not apparent at a larger scale and no difference in population density estimates could be identified between platforms. As noted above, detection probability depends on perception probability, estimated here, and the availability of targeted species. Availability can be considered consistent between platforms, but we demonstrated that the perception probability was platform dependent. As a consequence, we would expect that overall detection probability should be different between platforms. When estimating population densities with CDS, the detection probability can be incorporated by multiplying the estimate by this probability to obtain a density estimate corrected with detection. When multiplying the estimated densities by the perception probability identified here, we still obtained similar corrected density estimates between both platforms (not shown here). This indicates that the variation in overall perception probabilities is probably negligible compared with the uncertainty surrounding the estimates (confidence intervals). Therefore, the differences in perception probabilities between platforms do not impact the resulting population density estimates as obtained through CDS, and we are confident that the negative impact of seabird detections does not affect small cetacean population assessment derived from the Megafauna protocol.

The absence of bias in population estimates through CDS might further indicate that this effect could be negligible on cetacean detection in our study area compared with other parameters accounted for in distance sampling analyses, such as distance from the transect line, which is recognized as the main parameter affecting animal detection [4], or environmental observation conditions such as the Beaufort sea state (e.g. [14,15]). A more precise abundance estimation could have been obtained by considering observation conditions, such as Beaufort sea state or subjective conditions, using multi-covariate distance sampling (MCDS), but such analysis was precluded here by the reduced sample size for most studied species. It was tested on common/striped dolphin groups (the only species group with sufficient sample size), and resulted in similar density estimates to those from using CDS, so we are confident that using MCDS instead of CDS would not impact the conclusions presented here.

However, it must be kept in mind that the survey was conducted over temperate waters with moderate cetacean densities, and where seabirds and cetaceans show relatively little spatial overlap in their distributions [34]. It would be informative to run the double-platform protocol in the eastern English Channel and the North Sea, where there is stronger co-occurrence of seabirds and harbour porpoises at high densities [34]. For example, gulls and gannets can be present at such high densities that observers with experience in those areas reported their impression of a lower perception probability of harbour porpoises because the sea surface could at times be partially obscured by flying birds. In such areas, we might expect to find a negative effect of seabirds even for the Scans platform.

By contrast, a completely different result might be expected in tropical waters, where densities of both cetaceans and seabirds are far lower than in our study area [7]. In tropical waters, small cetaceans are more sparsely distributed and cetaceans and seabirds have opposing distributions [5]. With less co-occurrence, we might expect an absence of seabird detection effects on cetacean perception at the 30 s scale. However, observers with extensive aerial survey experience in tropical areas believe that detecting seabirds helps observers to stay focused and alert on the transect line. Indeed, in tropical waters, the sighting rate might be very low, with long sequences without any sighting. In such cases, recording seabirds might, in fact, be beneficial to the detection of cetaceans.

Previous exploration of the detection process in aerial surveys using double-platform protocols (but with the two platforms operating the same protocol, unlike here) have demonstrated variability in detection probabilities among observers [15,16,35,36], although in some cases this variability was shown to be negligible [37]. This variability might come from intrinsic differences in observer perception and experience, but also from what the observer actually sighted, and where. For example, a specific observer might have more experienced moderate observation conditions, or higher turbidity than another.

Observer effect has also been shown to directly impact population abundance estimates using distance sampling (see for example [38]). Here, we were able to disentangle whether the estimated

small cetacean densities were affected by observer-specific, platform-specific or species-specific bias in detection functions by implementing a hierarchical modelling of these detection functions. We found that there was no observer effect in ESW estimates. This analysis clearly demonstrated that the species effects overcome those of protocol and observers. Hence, we can conclude that there is no observer effect in population density estimated from CDS, and confirm the absence of platform effects on these density estimates.

In conclusion, in our study area, we found no evidence of lower performance regarding small cetacean population monitoring using a multi-target protocol compared with a mono-target protocol dedicated to cetaceans. The systematic recording of a wide range of marine taxa and anthropogenic objects following standardized protocols (strip or line transect) is both scientifically robust and cost-effective. As such, it facilitates parallel monitoring of the marine megafauna community, permitting the exploration of a large set of ecological questions at an ecosystem scale (e.g. [39]). The systematic recording of anthropogenic activities concurrently with the megafauna community also facilitates the assessment of anthropogenic threats faced by a wide range of species, and enables conservation strategies to be adjusted accordingly (e.g. [40]). Such a protocol represents a valuable option in the context of coordination and optimization of scientific surveys to achieve efficient and effective monitoring to restore or maintain biodiversity.

Data accessibility. Data and codes used to conduct the analyses are available on Dryad [41].

Authors' contributions. M.A., G.D., C.L., S.L., A.R., V.R., J.S. and O.V.C. conceived the ideas and designed the methodology; G.D., A.G., P.H., S.L., M.S. and O.V.C. collected the data; C.L. and M.A. processed the data and analysed the data; C.L. led the writing of the manuscript. All authors contributed critically to the draft and gave final approval for publication.

Competing interests. We declare we have no competing interests.

Funding. This work has been funded by the French ministry in charge of the environment (Ministére de la Transition Écologique et Solidaire).

Acknowledgements. We are deeply grateful to all the observers involved in the SCANS-III survey and to the leading team who organized the survey and made this experiment possible. We warmly thank the pilots and crew members of the PixAir Survey company. We are grateful to all the people involved in conducting and analysing data from the double-platform protocol implemented during pilot studies SAMM and SAMM-ME.

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
