## [Reviewer comments · Royal Society Open Science]

Review History

RSOS-190296.R0 (Original submission)

Review form: Reviewer 1 (Megan Ferguson)

Is the manuscript scientifically sound in its present form?

No

Are the interpretations and conclusions justified by the results?

Yes

Is the language acceptable?

No

Is it clear how to access all supporting data?

Yes

Do you have any ethical concerns with this paper?

No

Have you any concerns about statistical analyses in this paper?

Yes

Recommendation?

Major revision is needed (please make suggestions in comments)

Comments to the Author(s)

Please see attached comments (Appendix A).

Review form: Reviewer 2

Is the manuscript scientifically sound in its present form?

Yes

Are the interpretations and conclusions justified by the results?

Yes

Is the language acceptable?

Yes

Is it clear how to access all supporting data?

Yes

Do you have any ethical concerns with this paper?

No

Have you any concerns about statistical analyses in this paper?

No

Recommendation?

Accept with minor revision (please list in comments)

Comments to the Author(s)

Review of Lambert et al. "The effect of a multi-target protocol on cetacean detection and abundance estimation in aerial surveys".

Overall, it is a well written, fairly easy paper to follow, with some interesting and important conclusions that resulted from a well-developed analysis. The authors attempted to investigate many of the logical factors that could influence the results using a large dataset. Though the authors did acknowledge that given all the subsetting that was done, the dataset would have to larger to tease apart some issues.

Specific comments:

- 1) When estimating the perception probability why was distance from the track line not used? It seems a potentially very influential factor that is accounted for when using CDS, so why not in the perception probability estimation?
- 2) Line 156: it would be good to explain that 90 degrees is straight down not out to the horizon.
- 3) Line 260: did you say what x is in the ESW equation? And what is σ ?

4) Lines 378: this counterbalance seems important and influencing the overall conclusions. Seems accounting for the glare difference between the two platforms would be important and potentially shift your conclusions.

5) Lines 416-418: yes these would be very good statements to include. It seems you do have a large enough sample to do a MCDS analysis. But maybe not? Also why did you not try out a MRDS (mark recapture distance sampling) analysis with the two teams within the area and species overlapping between the two teams, even though they used two different procedures? This of course would not have addressed your primary questions, but would you of resulted in a better final abundance estimate.

6) Lines 419+: It seems the fact that seabirds and cetaceans showed relatively little spatial overlap in their distributions has a huge effect on the conclusions about seabirds. Perhaps to look at the effects of seabirds you have to only use the portions of the track lines where there was overlap. Given the reported impression in areas in high bird density areas in lines 423-426, you need to be much more cautionary about any conclusions relative to sea birds.

7) Lines 442-448: I'm not sure this paragraph is needed or else I missed the major point you wanted to make in it.

Decision letter (RSOS-190296.R0)

13-Jun-2019

Dear Dr Lambert,

The editors assigned to your paper ("The effect of a multi-target protocol on cetacean detection and abundance estimation in aerial surveys") have now received comments from reviewers. We would like you to revise your paper in accordance with the referee and Associate Editor suggestions which can be found below (not including confidential reports to the Editor). Please note this decision does not guarantee eventual acceptance.

Please submit a copy of your revised paper before 06-Jul-2019. Please note that the revision deadline will expire at 00.00am on this date. If we do not hear from you within this time then it will be assumed that the paper has been withdrawn. In exceptional circumstances, extensions may be possible if agreed with the Editorial Office in advance. We do not allow multiple rounds of revision so we urge you to make every effort to fully address all of the comments at this stage. If deemed necessary by the Editors, your manuscript will be sent back to one or more of the original reviewers for assessment. If the original reviewers are not available, we may invite new reviewers.

- Data accessibility

If you wish to submit your supporting data or code to Dryad (<http://datadryad.org/>), or modify your current submission to dryad, please use the following link:
<http://datadryad.org/submit?journalID=RSOS&manu=RSOS-190296>

- Competing interests

- Authors' contributions

- Acknowledgements

- Funding statement

on behalf of Dr Denise Greig (Associate Editor) and Kevin Padian (Subject Editor)
 openscience@royalsociety.org

Associate Editor's comments (Dr Denise Greig):

This is a very useful study which will be helpful for standardizing multi-target aerial survey protocols across international boundaries and conservation initiatives. Both reviewers are supportive and provide many ideas for improving your manuscript. The only additional comment from me is that you may want to reconsider the colors used in your figures. They look the same when your manuscript is printed in black and white and I suspect they would be difficult for a color blind reader to differentiate.

Comments to Author:

Reviewers' Comments to Author:
 Reviewer: 1

Comments to the Author(s)
 Please see attached comments.

Reviewer: 2

Comments to the Author(s)
 Review of Lambert et al. "The effect of a multi-target protocol on cetacean detection and abundance estimation in aerial surveys" .

Overall, it is a well written, fairly easy paper to follow, with some interesting and important conclusions that resulted from a well-developed analysis. The authors attempted to investigate many of the logical factors that could influence the results using a large dataset. Though the authors did acknowledge that given all the subsetting that was done, the dataset would have to larger to tease apart some issues.

Specific comments:

- 1) When estimating the perception probability why was distance from the track line not used? It seems a potentially very influential factor that is accounted for when using CDS, so why not in the perception probability estimation?
- 2) Line 156: it would be good to explain that 90 degrees is straight down not out to the horizon.
- 3) Line 260: did you say what x is in the ESW equation? And what is σ ?
- 4) Lines 378: this counterbalance seems important and influencing the overall conclusions. Seems accounting for the glare difference between the two platforms would be important and potentially shift your conclusions.

5) Lines 416-418: yes these would be very good statements to include. It seems you do have a large enough sample to do a MCDS analysis. But maybe not? Also why did you not try out a MRDS (mark recapture distance sampling) analysis with the two teams within the area and species overlapping between the two teams, even though they used two different procedures? This of course would not have addressed your primary questions, but would you of resulted in a better final abundance estimate.

6) Lines 419+: It seems the fact that seabirds and cetaceans showed relatively little spatial overlap in their distributions has a huge effect on the conclusions about seabirds. Perhaps to look at the effects of seabirds you have to only use the portions of the track lines where there was overlap. Given the reported impression in areas in high bird density areas in lines 423-426, you need to be much more cautionary about any conclusions relative to sea birds.

7) Lines 442-448: I'm not sure this paragraph is needed or else I missed the major point you wanted to make in it.

Author's Response to Decision Letter for (RSOS-190296.R0)

See Appendix B.

RSOS-190296.R1 (Revision)

Review form: Reviewer 2

Is the manuscript scientifically sound in its present form?

Yes

Are the interpretations and conclusions justified by the results?

Yes

Is the language acceptable?

Yes

Do you have any ethical concerns with this paper?

No

Have you any concerns about statistical analyses in this paper?

No

Recommendation?

Accept as is

Comments to the Author(s)

Authors seem to have addressed the reviewers' comments. More data would have been nice to do more through analyses using standard methods, but what was done seems sufficient to publish.

Decision letter (RSOS-190296.R1)

09-Aug-2019

Dear Dr Lambert,

I am pleased to inform you that your manuscript entitled "The effect of a multi-target protocol on cetacean detection and abundance estimation in aerial surveys" is now accepted for publication in Royal Society Open Science.

Best regards,

on behalf of Dr Denise Greig (Associate Editor) and Kevin Padian (Subject Editor)
openscience@royalsociety.org

Associate Editor Comments to Author (Dr Denise Greig):

Thank you for your thoughtful comments to the reviewers and improvements to the manuscript.

I just saw a few typos:

line 27, change "tp" to "to"?

line 432, either replace "in a" with "using", or add "analysis" after (MCDS)?

line 457-458. Are you saying "a specific observer might have more experience with moderate observation conditions, or higher turbidity, than another"?

Thanks again - this paper will be very useful for making decisions about the most appropriate cetacean detection protocol.

Reviewer comments to Author:

Reviewer: 2

Comments to the Author(s)

Authors seem to have addressed the reviewers' comments. More data would have been nice to do more through analyses using standard methods, but what was done seems sufficient to publish.

Appendix A

Review of Lambert et al.

Manuscript ID RSOS-190296

General Comments

Lambert et al. investigate whether the perception probability and resulting density or abundance estimates of small cetaceans during aerial line-transect surveys are affected by simultaneously collecting different types of data on non-target objects. This is an unexplored topic and would be a valuable addition to the distance sampling literature. In general, the survey and analytical methods seem well suited to address the question, although below I detail specific concerns that need to be addressed. The authors' conclusions logically follow from their results. The paper would be much easier to read if considerable structural and stylistic edits were made, as noted in my detailed comments below.

I recommend that the authors address my analytical concerns and revise the text and figures to be more reader-friendly. I would be happy to review a revised draft of the manuscript.

Detailed Comments

Notes: In my notation below, L### refers to the running line number. Overarching comments do not have an associated line number. When I recommend different wording, I often just list my recommended revision, I often do not also list the original wording.

1. Be consistent about whether SCANS-III is hyphenated (most of the text) or not (first line in the abstract)
2. [abstract, L1]: “during the SCANS-III survey”
3. [abstract]: The following sentence confused me when I first read the paper: “Small cetacean perception was higher following the detection of another cetacean within previous 30 seconds in both platforms, but decreased following the recording of any seabirds in the Megafauna platform only.” I read just a few sentences prior that only the Megafauna platform recorded seabirds, so it seemed logical that recording seabirds could only affect the Megafauna data. How could recording seabirds affect the platform in which seabirds were not recorded? Could this be restated as, “Small cetacean perception was higher following the detection of another cetacean within previous 30 seconds in both platforms. The only prior target that decreased small cetacean perception during the subsequent 30-sec period was seabirds, in the Megafauna platform.”
4. In the abstract, you refer to “a small temporal scale (30 seconds” in contrast to “a larger scale (>10 km).” The reader is left wondering how far your survey aircraft can go in 30 seconds, and how that compares to 10 km. I did the math: you travel ~1.391667 km in 30 sec at 167 km/hr. Use common units to define small scale and large scale. Additionally, why are you defining large scale to be >10 km? In the manuscript, I understood your large-scale metric to be your overall abundance estimate, which was at the scale of your study area. Once this issue of scale is addressed, I recommend you edit this sentence as follows: “However, at a larger scale (###), this small-scale perception bias had no effect on the density estimates, which were similar for the two protocols.” Density estimation is the process of estimating density. One of the key metrics you are interested in is the density estimate itself.
5. [abstract] Because our study area was characterized by moderate cetacean densities and small spatial overlap of cetaceans and seabirds, any extrapolation to other areas or times requires caution.
6. [abstract, last sentence]: “..., the multi-target protocol is valuable for optimizing logistical and financial resources to efficiently monitor biodiversity and study community ecology.”

7. The first paragraph of your Intro is mostly about policy. When I read it, I expected a policy-dominated manuscript. The opening to your Intro should be strong and it should signal to the reader where you will be taking them during the rest of the story. The length of the opening should be adjusted to suit the audience's background. You are submitting this to a journal with an audience that has a very broad scientific background, so your opening can be as long as an entire paragraph. I recommend deleting your entire first paragraph. The paragraph spanning L12-17 is a much better opener. I'd keep the first sentence (L12) as is. Then, I'd focus on the trade-offs associated with resources (fuel, logistics, cost, qualified personnel) required to conduct multiple surveys versus the potential detrimental effects to the accuracy, precision, and bias (and our ability to estimate each) of the resulting density estimate for the target species/taxon. I would also allude to the preference for ecosystem- (or community-) based monitoring and conservation.
8. [L13] The sentence beginning "The need for reliable estimates has led to..." is a throw-away sentence in my opinion. It doesn't provide any information about what specific topics were addressed, and it doesn't shape your story at all. Save a few words in your word count and save the reader from reading this sentence. Just delete it!
9. [L19] Recommended edit: "In a marine context, aerial surveys have an advantage over boat-based surveys by allowing large areas to be covered in a relatively short period, maximizing the ability to collect data during transient good weather conditions. Vessels, however, may have a larger offshore range due to their low fuel consumption rates relative to fuel storage capacity." However, I think you can delete this paragraph entirely because it's the only place in the entire manuscript where you discuss boat-based surveys. The paper has nothing to do with vessel vs. aerial surveys. Focus on the primary issue of whether collecting ancillary data on non-target objects negatively affects inference on the target object.
10. [L24]: Strip transect sampling assumes perfect detection
11. [L29 & other places] Availability bias is a function of the objects surface duration & dive duration, and also the amount of time a patch of water is in the observer's field of view. Time in view is a function of altitude, speed, and window configuration.
12. [L31] Recommended edit: "... (only visible sub-surface) ...". However, what exactly do you mean by only visible subsurface? Do sharks never show their body parts above the surface of the water? "Subsurface" itself is very vague because the bottom of the ocean is technically subsurface, but an aerial observer will not see a shark resting on the bottom of the ocean.
13. [L33-34] Here, you state, "Mono-target protocols maximize the data quality and minimize the uncertainties around distribution and abundance estimate for the targeted species...." If this were universally true, you would have no reason to write your paper. I would change this to something like, "Mono-target protocols are conventionally thought to maximize...."
14. [L38] In the context of optimizing marine wildlife monitoring, multi-target surveys have several advantages, including cost-sharing, reducing the ecological footprint compared
15. [L41] "...ecological relationships among ecosystem components."
16. [L42] This line refers to "relevant monitoring programmes." What do you mean by "relevant"? Are you talking about ecosystem-based monitoring programs?
17. [L44] However, a possible drawback to implementing multi-target protocols is...
18. [L49] "...thus the observer's capacity to detect an animal present at or just beneath the surface might be reduced, with a corresponding increase in perception bias." Alternatively, you could state "corresponding decrease in perception probability".
19. [L51] mono-target protocols has
20. [L52] ...and availability, among other factors
21. [L53] ...but the effect of detecting targets with different characteristics
22. [L55] ...on their abundance estimates has so far

23. [L56] types of protocols
24. [L57] Need a comma between “or aircraft” and “and combination” to reduce ambiguity here.
25. [L60] ...of two independent observer teams
26. [L62-67] I was confused for quite a while about what the real difference was between the data collection protocols during SCANS-III and Megafauna. Both collect data on cetaceans, anthropogenic objects, turtles, sharks, nets, and litter. The only taxon that is different between the two is seabirds. At this point in the paper, I would simplify things by saying something like, “The Scans platform collected a full suite of line-transect data on the target species, small cetaceans, but collected a limited subset of data on non-target objects (boats, nets, marine litter, fish, and jellyfish). The Megafauna platform collected line-transect data on small cetaceans and non-target objects, and included flying or resting seabirds in the list of recorded objects.”
27. [L68/69] When I got to this point in the paper, I realized that you seem to use detection and perception interchangeably. I think it would be good to take a close read and make sure that the distinction is clear and consistent throughout the ms. The terms availability bias and perception bias are second-nature to people who are familiar with distance sampling, but you are submitting this to a journal with a broad readership, so you should do the best you can to simplify the mental load the reader needs to use in order to understand the terminology.
28. [L81] ...platform of observation on the perception process for small cetaceans.
29. General question about your data collection methods: how were the data recorded? My guess, after reading between the lines, is that each observer team was comprised of 3 observers, with 2 people as designated observers and a third person as a data recorder who entered data in real-time into a laptop. Is this correct? These details are critical to understanding your results. One of the issues likely affecting perception bias in multi-target studies is just the burden of recording data. You could even elaborate on this more in the intro: complications arising from multi-target studies include the issue of search image (the observer isn't tuned into a single search image), observer workload (collecting the full suite of data on multiple target types, which all may be available at the same time), and data recorder workload (if the observers have to relay data to the recorder, they can only do it as fast as the recorder can input information).
30. [L85] ...Norway, to the Strait
31. [L91] “...data recording) and between platforms.” As a reader, I'd much rather read one single word (“and”) rather than three words (“as well as”) if information isn't lost.
32. [L98] Does swell refer to swell height or direction?
33. Need to be consistent throughout the ms regarding whether you hyphenate “line-transect” and “strip-transect”.
34. General question: did you ever encounter or record pinnipeds?
35. [L112 and 114] “fishing gear” because “gear” is plural already
36. [L116] and jellyfish, but detection angle
37. [L117] focused
38. [L118] ...swim direction, number of animals, presence of calves, and any reactions of the animals to the aircraft were recorded. Seabird activity (resting or fling) was also recorded.
39. [L120-121] I'd remove the parentheses from this sentence because the stuff in the parens is important to the sentence; enclosing it in parens subverts its importance.
40. [L124] ...of survey effort, because the platforms were independent.
41. [L125] The majority of the survey effort (84% and 74% for Megafauna and Scans platforms, respectively) was conducted...
42. [L127] ...information only, pooling all small cetacean sightings recorded on effort: ...
43. I interpret your methods to mean that you included unidentified cetaceans in your “all marine mammals” analyses to estimate ESW and compute density. If so, this violates the fundamental

assumption of distance sampling that objects are distributed randomly with respect to the transect. Unid cetaceans cannot be treated like cetaceans identified to species because one would expect their density to be greater farther from the transect, especially for a survey operating in passing mode. It's harder to identify things farther away, so you're going to have more unids at greater distances from the trackline. Have you generated a histogram of perpendicular sighting distances for your unids? My guess is that it doesn't look anything like a half-normal or hazard-rate function. Unid cetaceans should not be included in analyses to estimate ESW, and they should not be considered equivalent to sightings identified to species in order to estimate encounter rate or average group size in a distance sampling analysis. I discussed this issue with Jeff Laake several years ago and wrote up his recommendations, which I've copied to the end of this document.

44. [L130-131] I don't understand how you defined your sample units in order to estimate the variance in your encounter rate and density estimators. Furthermore, it isn't clear what variance estimator you used. I think that the default estimator in package Distance is Fewster's R2 estimator, which was based on the idea that the sample units correspond to transect segments. Effort and sightings from transects flown repeatedly within a short period of time should be pooled into the same sample unit.
45. [L138] "Segments were then paired...." What is meant by "segments" here?
46. [Section 2.4.1] Here and in Figure 3, you need to clarify that the 500-m buffer around the focal sighting is omnidirectional (it's a circle), whereas the 15° interval is only perpendicular to the transect line and that "15°" refers to the angle of declination from the horizon to the sighting.
47. How were declination angles measured? I am interpreting 15° to be your accepted measurement error at 600 ft altitude. Quick math tells me that the perpendicular distance between a declination of 90 deg vs. 75 deg at 600 ft alt is approximately 50 m, whereas the distance between 75 deg and 60 deg is approximately 60-ish m. I think it might help the reader understand your matching criteria if you could insert some helpful insights into the linear distances (double-check my math!) corresponding to the declination parameter. I also think it's worthwhile to somewhere state that you recognize that measurement error exists in declination angles and that is one reason for needing to implement this hierarchical method for finding duplicates.
48. [L169-172] Here you state that availability bias was assumed constant across platforms. As I noted above, this is true only if objects had the same time-in-view for both platforms. Were the windows the same size and shape? Were the fields of view identical? This needs to be stated explicitly to back up the claim/justify the assumption that the availability bias cancels out.
49. [L185] Not sure if it's worth noting, but I think the 30-sec threshold might need to be adjusted in other situations where animals might be found in very large aggregations (e.g, groups of 3000 belugas) or where clusters of 1-10 individuals might come at you rapid-fire in a foraging area. If one of your readers is surveying in one of those cases, they shouldn't take your 30-sec threshold to be universally good!
50. [L184] remove the comma after "covariates"
51. [L198-206] I'd move this entire paragraph to be the very last paragraph of your intro. This lists your hypotheses – it's helping to define the challenge you face as you work through the story that you present in the paper. This should come right after your goals ("aim") and objectives.
52. [L210]: ...angle was less than 5° (a more restrictive tree) or 25° (a less restrictive tree).
53. [L219] conditions (sea state < 4 and moderate to excellent subjective conditions) were included in the analysis
54. [L226] what model selection criteria did you use to select the half-normal? I think that should be stated in methods. I think you should move the statement about selecting the half-normal to the results section
55. [L227] "effective strip width" does not need to be capitalized, even though the abbreviation is ESW

56. [Section 2.5.2] Encounter rates of cetaceans could be correlated with “class” if there are mixed species hotspots or observation effects (e.g., sea state, low clouds, fog) that affect perception bias. Another question for this section: what was the sample unit for estimating encounter rate and its associated variance in this analysis? If you based it on something smaller than a single transect line, what did you do to investigate and address, if necessary, spatial autocorrelation?
57. [L250] I don’t understand why you needed to segment the data for the hierarchical model. If I understand your model correctly, it takes point data as the input.
58. [L252-253] What is meant by “shrunk towards a common detection function”?
59. [Section 2.5.3] I’m having trouble understanding your model structure here. I understand that you’re ‘using Bayesian methods, so I expect to see definitions for a likelihood function and prior distribution(s), along with possibly some hyperpriors. Is a “semi-normal” distribution the same thing as a “half-normal” distribution? Your data are the perpendicular distances, so those belong in the detection function, which I think you’ve defined to be “semi-normal.” How did you define the mean and variance of the semi-normal? I think the variance is defined by your σ_{jkl} . If so, that means that your δ , ϵ , and γ each need a prior distribution and probably some hyperpriors. Why did you model the random effects using a bivariate distribution? Are you assuming that observer and species effects might be correlated? I got really, really confused here. Lastly, why did you choose to use Bayesian inference here? My understanding is that there wasn’t a considerable amount of prior information that could be used to inform the priors. Why go to the hassle of a Bayesian analysis if the rest of the paper was frequentist and you don’t have a pool of relevant data for developing priors?
60. [L259] “centered” is misspelled
61. [L264] and 0.45 sightings per km for
62. [L265] over both platforms (0.07 sightings per km for each target type for the Megafauna platform; 0.07 and 0.06 sightings per km, respectively, for the Scans platform). The seabird encounter rate....
63. [L272] I didn’t understand the following sentence: “Marine mammals and other marine fauna had a null mean difference, while mean difference for anthropogenic objects was slightly greater for the Scans platform.”
64. [L279] The difference in small cetacean perception probability p between the two platforms in the absence of previous target detection....
65. [L281] of previous targets on p were significant, ...
66. [L291] while detections of resting seabirds and seabirds in groups increased....
67. [L309] This states, “CDS-based ESW estimation indicated a platform-independent ESW for all taxonomic groups....” However, the fact that the confidence intervals overlap almost entirely in Figure 7 doesn’t seem to support this statement.
68. [L314] This line says “two platforms for the three taxonomic groups” but I count 4 groups in Figure 7.
69. [L334] It is possible that this protocol could result in biased or imprecise population estimates for small cetaceans if recording other taxa with different detection characteristics introduces a bias in small cetacean detectability.
70. [L340] The amount of effort and sightings with the double-platform protocol during SCANS-III (>10,400 km and >700 cetacean sightings) was far greater than for previous surveys (>2000 km and < 200 cetaceans), allowing us to identify clear patterns in small cetacean detections within and between the two tested protocols.
71. [L344] Delete “As explained above” from the start of this sentence
72. [L344-345] perception probability of small cetaceans because availability bias was assumed constant across platforms.

73. [L346] Delete the parentheses because the stuff inside the parens is worthy of being part of the main idea of the sentence
74. [L347] It might originate in the observer focusing better on the transect while operating the Megafauna protocol, as shown by an overall higher detection rate. [I think you can omit the stuff in parens because that point should be obvious to the reader by now. I'd also delete the following sentence, beginning, "However, such a higher detection rate...", because this paragraph is trying to explain the apparent *unintuitive* result.]
75. [L353] conditions were considered more comfortable
76. [L354] (Scans) which was exposed to direct glare
77. [L354] You state, "Unfortunately, we were not able to directly test this aspect." I would change "aspect" to "difference." Also, this could easily be tested in future field efforts by just swapping the location of the teams using each survey protocol.
78. [L355-359] I got a little confused by this paragraph. I think you're just trying to say that your perception probabilities were consistent with those estimated for other surveys. I would trim this down to a single sentence and move it right before the sentence beginning "Our results highlighted a slightly higher..." that is currently on L345.
79. [L363] Omit the sentence, "Again, we cannot determine the reasons behind it." You've already stated this, and there's no reason to dwell on ignorance! You should focus on your achievements.
80. [L364-368] Here you're hypothesizing why cetacean presence in the previous 30 sec induced a higher perception probability for both platforms. You suggest that the result is consistent with what could result from spatially heterogeneous cetacean density in your study area. However, I don't think that high density patches alone can explain the higher *perception* probability. Areas of high density could very likely explain the probability that another cetacean is encountered in the next 30 sec, but that is different than perception probability (i.e., $p(\text{detect} | \text{encounter})$).
81. [L369] We use seabirds as a cue all the time for marine mammal aerial surveys conducted in the Arctic. Sometimes gray whales that feed benthically stir up prey that seabirds can reach from the surface. Sometimes the cetaceans and seabirds are all feeding on a near-surface bait ball. I think the effect of allowing an observer's gaze to stop momentarily on a seabird could easily increase or decrease perception probability.
82. [L374] but not from masking the sea surface, in which case... [remove the parentheses in order to bring the stuff in parens on equal footing with the rest of the sentence
83. [L375] cetaceans; seabirds were mostly inshore, whereas cetaceans were mostly offshore (Lambert et al. 2017).
84. [L377] ...negligible impact on high-density cetacean areas.
85. [L379] probability for this platform, but would affect spatially explicit models and inference into community ecology.
86. [L380] We expected to find that detection of other marine fauna (large fish, turtles, jellyfish) would have a positive impact on small cetacean perception probabilities for both platforms. However, this was true only for the Megafauna platform; for the Scans platform, we found no effect.
87. [L390] regarding the beneficial effects of seabirds
88. [L400-408] This section confused me a little. My biggest concern was that the CDS density estimator says pretty clearly that, all other things being equal, a decrease in perception probability should result in a higher density estimate. Therefore, the fact that your density estimates were comparable even though you detected slight differences in perception probabilities leads me to believe that there must have been platform-specific differences in encounter rate or group size estimates. Based on the math, something needs to compensate for the differences in p.
89. [L409-415] I found this paragraph to be very repetitive of information said multiple times previously. I'd delete it.

90. [L415-418] I agree with your concern here that an mcds analysis would be better. This gets back to my comment about section 2.5.2 (item #56 in the list above). Could you implement mcds just for something like sea state? I find it really hard to believe that sea states in the range of Beaufort 0 to 4 have no effect on detectability of small cetaceans, even at 600 ft survey altitude! You could simplify things by pooling Beaufort 0-2 and 3-4, thereby creating only 2 categories.
91. [L442-448] I disagree with the statement that you'd need to have a given observer operating each protocol over the exact same configuration of sm cetacean sightings in order to investigate perception probability at the observer level. If there is a non-trivial effect, I think you'd just need a good sample size and you could evaluate this statistically. Or create a video game and see how distracted they get when you ask them to do multi-species tasks as opposed to focusing on a single species!
92. Figure 1: Insert a scale bar.
93. Figure 2: This was VERY helpful to me!
94. Figure 3b: The purpose of this figure is to help the reader visualize the matching algorithm. The matching algorithm addresses a 3-Dimensional problem (time plus space in 2D), but you can get away with illustrating just the 2D spatial components. This figure seems to plot all sightings in 1D – they are all plotted at the point on the transect line perpendicular to the actual sighting. I think this figure would be much more intuitive if you plotted the sightings perpendicular to the transect at the location specified by their respective declination angles.
95. Figures 5-9: Need to clarify what the error bars represent. 95% CIs?

Adjusting for Species-Identification Bias

The problem: During line-transect aerial surveys for marine mammals, observers are unable to identify all sightings to species.

Distance sampling theory says: “Observers should be allowed to record a detection as an unknown species, and they should be trained to focus their effort on identifying species for detections close to the line or point. Unidentified observations away from the line can be excluded from the analysis, effectively treating them as if they had been undetected.” (Buckland et al., 2001 pg. 302)

How to correct for unids sighted close to the line: If sightings of unids are close to the line and detectability is similar across all species i in the unid category, can correct for species-identification bias as follows.

1. Compute the probability of recording a unid species within the strip (dx) in which detection is assumed to be the same for all species. $n_{unid, dx}$ = number of unid sightings within strip dx . $n_{i, dx}$ = number of sightings identified to species i that were located within dx .

$$P(unid) = \frac{n_{unid, dx}}{n_{unid, dx} + \sum_i n_{i, dx}}$$

2. The probability of identifying a sighting to species is

$$P(ID) = 1 - P(unid)$$

3. To correct for species-identification bias, divide estimated density of species i ID'd to species by $P(ID)$

$$\hat{D}_i^* = \frac{\hat{D}_i}{P(ID)}$$

where \hat{D}_i^* is the adjusted estimated density of species i after correcting for species-ID bias.

- Use the same estimate of $P(ID)$ to correct estimated density for all i species
4. After the unid sightings have been used to compute $P(ID)$, exclude them from the rest of the distance sampling analysis

Literature Cited

Buckland, S.T., D.R. Anderson, K.P. Burnham, J.L. Laake, D.L. Borchers, and L. Thomas. 2001. Introduction to distance sampling: estimating abundance of biological populations. Oxford University Press, Oxford. 432 pages.

Appendix B

Ref: RSOS-190296

Title: The effect of a multi-target protocol on cetacean detection and abundance estimation in aerial surveys

Journal: Royal Society Open Science

Dear Editor,

Thank you for your interest regarding our manuscript. We would like to warmly thank the two reviewers for their thorough review of the manuscript, which clearly improved the paper. We followed your recommendation to modify colours used in figures, and modified it in a way that could be read when printed in grey scale. All suggestions made by reviewer 1 for rewording were followed. All other comments from reviewer 1 and all comments from reviewer 2 are presented and answered hereafter, detailing the modifications made in the text. In black are the reviewers' comments, and in red our answer. We provided a pdf from Latex for the revised manuscript, so we did not have file tracking the changes. We instead indicated the lines where the changes can be found in the revised manuscript.

Thanks again for your interest,

Sincerely,

Charlotte Lambert

Reviewer 1

[abstract]: The following sentence confused me when I first read the paper: "Small cetacean perception was higher following the detection of another cetacean within previous 30 seconds in both platforms, but decreased following the recording of any seabirds in the Megafauna platform only." I read just a few sentences prior that only the Megafauna platform recorded seabirds, so it seemed logical that recording seabirds could only affect the Megafauna data. How could recording seabirds affect the platform in which seabirds were not recorded? Could this be restated as, "Small cetacean perception was higher following the detection of another cetacean within previous 30 seconds in both platforms. The only prior target that decreased small cetacean perception during the subsequent 30-sec period was seabirds, in the Megafauna platform."

We modified the text following the suggestion: "Small cetacean perception was higher following the detection of another cetacean within the previous 30 seconds in both platforms. The only prior target that decreased small cetacean perception during the subsequent 30 seconds was seabirds, in the Megafauna platform".

In the abstract, you refer to "a small temporal scale (30 seconds" in contrast to "a larger scale (>10 km)." The reader is left wondering how far your survey aircraft can go in 30 seconds, and how that compares to 10 km. I did the math: you travel ~1.391667 km in 30 sec at 167 km/hr. Use common units to define small scale and large scale. Additionally, why are you defining large scale to be >10 km? In the manuscript, I understood your large-scale metric to be your overall abundance estimate, which was

at the scale of your study area. Once this issue of scale is addressed, I recommend you edit this sentence as follows: "However, at a larger scale (###), this small-scale perception bias had no effect on the density estimates, which were similar for the two protocols." Density estimation is the process of estimating density. One of the key metrics you are interested in is the density estimate itself.

*We followed the suggestions and modified the two sentences as follows: "At a small temporal scale (30 seconds, **roughly 1.5 km**), our results provided overall similar perception probabilities for both platforms. Small cetacean perception was higher following the detection of another cetacean within the previous 30 seconds in both platforms. The only prior target that decreased small cetacean perception during the subsequent 30 seconds was seabirds, in the Megafauna platform. However, at a larger scale (**study area**), this small-scale perception bias had no effect on the density **estimates, which were similar for the two protocols.**"*

The first paragraph of your Intro is mostly about policy. When I read it, I expected a policy-dominated manuscript. The opening to your Intro should be strong and it should signal to the reader where you will be taking them during the rest of the story. The length of the opening should be adjusted to suit the audience's background. You are submitting this to a journal with an audience that has a very broad scientific background, so your opening can be as long as an entire paragraph. I recommend deleting your entire first paragraph. The paragraph spanning L12-17 is a much better opener. I'd keep the first sentence (L12) as is. Then, I'd focus on the trade-offs associated with resources (fuel, logistics, cost, qualified personnel) required to conduct multiple surveys versus the potential detrimental effects to the accuracy, precision, and bias (and our ability to estimate each) of the resulting density estimate for the target species/taxon. I would also allude to the preference for ecosystem- (or community-) based monitoring and conservation.

As suggested, we modified the opening of the intro to make it start at the second paragraph, and subsequently reshaped the first paragraphs as suggested (Lines 2-42). We think the introduction now flows better.

[L13] The sentence beginning "The need for reliable estimates has led to..." is a throw-away sentence in my opinion. It doesn't provide any information about what specific topics were addressed, and it doesn't shape your story at all. Save a few words in your word count and save the reader from reading this sentence. Just delete it!

Deleted in this new intro.

[L19] Recommended edit: "In a marine context, aerial surveys have an advantage over boat-based surveys by allowing large areas to be covered in a relatively short period, maximizing the ability to collect data during transient good weather conditions. Vessels, however, may have a larger offshore range due to their low fuel consumption rates relative to fuel storage capacity." However, I think you can delete this paragraph entirely because it's the only place in the entire manuscript where you discuss boat-based surveys. The paper has nothing to do with vessel vs. aerial surveys. Focus on the primary issue of whether collecting ancillary data on non-target objects negatively affects inference on the target object.

As suggested, we removed any reference to boat surveys and focused on the potential effect of detecting multiple targets on their respective detection probabilities.

[L62-67] I was confused for quite a while about what the real difference was between the data collection protocols during SCANS-III and Megafauna. Both collect data on cetaceans, anthropogenic objects, turtles, sharks, nets, and litter. The only taxon that is different between the two is seabirds. At this point in the paper, I would simplify things by saying something like, "The Scans platform collected a full suite of line-transect data on the target species, small cetaceans, but collected a limited subset of data on non-target objects (boats, nets, marine litter, fish, and jellyfish). The Megafauna platform collected line-transect data on small cetaceans and non-target objects, and included flying or resting seabirds in the list of recorded objects."

We modified the text as follows (Lines 50-56): The "Scans" platform used the same dedicated protocol as on all other survey aircraft in the SCANS survey, **and collected line-transect data on cetaceans but also recorded presence of non-target objects** (boats, nets, marine litter, fish, turtles and jellyfish; Hammond et al., 2002; Hammond et al., 2013). The "Megafauna" platform used a multi-target protocol **to collect line-transect and strip-transect data on a full suite of target objects, from jellyfish to cetaceans and seabirds** (same protocol as for REMMOA, SAMP and ObSERVE surveys; Manno et al., 2014a; Pettex et al., 2014; Laran et al., 2017; Rogan et al., 2018).

[L68/69] When I got to this point in the paper, I realized that you seem to use detection and perception interchangeably. I think it would be good to take a close read and make sure that the distinction is clear and consistent throughout the ms. The terms availability bias and perception bias are second-nature to people who are familiar with distance sampling, but you are submitting this to a journal with a broad readership, so you should do the best you can to simplify the mental load the reader needs to use in order to understand the terminology.

The manuscript was checked carefully to be sure not to use the terms interchangeably. We simplified the sentence as (Lines 57-58): *"This configuration allowed exploration of the effect of surveying different targets simultaneously on the perception of small cetaceans, given they were known to be there, and on the resulting abundance estimates."*

[L81] ...platform of observation on the perception process for small cetaceans.

The hierarchical modelling does not refer to perception process, but aims to disentangle the effect of protocol, observer and species in the observed ESW. We modified the text as follows (Lines 75-76): *"Finally, we implemented a hierarchical modelling of detection functions to disentangle the confounding effects of species, observer and platform of observation on the way small cetaceans are detected."*

General question about your data collection methods: how were the data recorded? My guess, after reading between the lines, is that each observer team was comprised of 3 observers, with 2 people as designated observers and a third person as a data recorder who entered data in real-time into a laptop. Is this correct? These details are critical to understanding your results. One of the issues likely affecting perception bias in multi-target studies is just the burden of recording data. You could even elaborate

on this more in the intro: complications arising from multi-target studies include the issue of search image (the observer isn't tuned into a single search image), observer workload (collecting the full suite of data on multiple target types, which all may be available at the same time), and data recorder workload (if the observers have to relay data to the recorder, they can only do it as fast as the recorder can input information).

As explained in the ms, each team was composed of 3 persons who switched between observation and data recording at regular intervals. In addition to these rotations, each team switched protocols.

The two protocols had a system of audio recording, ensuring that information given by observers was not lost in case the data recorder had no time to input all the information (in particular when a large amount of targets were detected at the same time). We added the following text (Lines 96-100): *"The two protocols had a system of audio recording, ensuring that information given by observers was not lost when the number of targets detected at the same time was too large to permit real-time recording of the whole suite of information by the data recorder. In such cases, potentially unrecorded in-situ sightings were reconstructed afterwards based on GPS and audio tracks."*

A few words regarding the complications due to multi-target and observation workload have been added to the intro (Lines 23-25).

[L98] Does swell refer to swell height or direction?

Neither height nor direction, swell refers only to the presence of swell potentially interfering with detections. It is coded as follows: 0 no swell; 1 swell, but not interfering; 2 interfering swell. We added details in the Methods section (Lines 93): *"Whether swell presence impacted detection was also recorded by the Megafauna platform"*.

General question: did you ever encounter or record pinnipeds?

There were just a few encounters, something like 2-3 individuals, in the study area. In general, pinnipeds are rather rarely seen at sea during aerial surveys. For example, they are rare in surveys of French waters, but are more frequently sighted on beaches at haul out sites during surveys in the eastern English Channel.

I interpret your methods to mean that you included unidentified cetaceans in your "all marine mammals" analyses to estimate ESW and compute density. If so, this violates the fundamental assumption of distance sampling that objects are distributed randomly with respect to the transect. Unid cetaceans cannot be treated like cetaceans identified to species because one would expect their density to be greater farther from the transect, especially for a survey operating in passing mode. It's harder to identify things farther away, so you're going to have more unids at greater distances from the trackline. Have you generated a histogram of perpendicular sighting distances for your unids? My guess is that it doesn't look anything like a half-normal or hazard-rate function. Unid cetaceans should not be included in analyses to estimate ESW, and they should not be considered equivalent to sightings identified to species in order to estimate encounter rate or average group size in a distance sampling analysis. I discussed this issue with Jeff Laake several years ago and wrote up his recommendations, which I've copied to the end of this document.

You understood correctly, individuals not identified to species level were introduced into the CDS analysis. As detailed in the ms, the group "all marine mammals" incorporates small and large cetaceans. However, the few sightings of "large unidentified cetaceans", which were indeed far from the line, were all removed before all the analysis presented in this paper. Large cetaceans considered for analysis were identified as pilot whales, beaked whales or rorquals. The "unidentified" groups included in the current analysis were identified as "small delphinids" (either striped, common or bottlenose dolphin for the study area), or as "common or striped dolphins". Importantly, the histogram of perpendicular distances of these individuals were almost identical to that for identified individuals. This reflects the fact that it is not always easy to identify these dolphin species from aerial survey even close to the transect line. Therefore, we believe incorporating unidentified groups into our analysis has permitted us to maximise our sample size and get more precise estimates, without having introduced a bias. Nevertheless, thank you very much for the useful note you copied to your review, it will surely be very useful for further analyses!

[L130-131] I don't understand how you defined your sample units in order to estimate the variance in your encounter rate and density estimators. Furthermore, it isn't clear what variance estimator you used. I think that the default estimator in package Distance is Fewster's R2 estimator, which was based on the idea that the sample units correspond to transect segments. Effort and sightings from transects flown repeatedly within a short period of time should be pooled into the same sample unit.

Encounter rates presented here are simple descriptive statistics: they were computed as the ratio of the sum of detections divided by the sum of effort within the surveyed region (see next comment). We did not compute or report associated standard errors. We chose to do so because the focus of our study was on the differences between the two protocols given that a survey will be carried out. The main source of variance in encounter rates is that of the placement of transects; the variance resulting from the design of the survey will affect estimates similarly for both platform.

The sampling unit used for density estimation was the leg of homogeneous detection conditions, as specified in the section 2.5.1 along with detailed CDS procedure. Fewster's R2 variance estimator was used – it is the only variance estimator currently available within the distance package, although alternative estimators are available in the mrds package.

[L138] "Segments were then paired...." What is meant by "segments" here?

Thank you for having spotted this oversight. We forgot to define what are "segments". We rephrased most of the paragraph to clarify the analysis made here, which is purely descriptive and based on 10 km long segments, unlike the rest of the analysis. The title of sections 2.3 and 3.1 were changed from "Encounter rates" to "Descriptive statistics", and the text was modified as follows (Lines 135-141): *"Sightings were classified into four main categories of interest for this study: anthropogenic objects (marine litter, fishing gear), marine mammals (mostly cetaceans), other marine fauna (not marine mammals) and seabirds. **Encounter rates were computed for each platform and category as the ratio of the total number of sightings over the sum of effort within the study area. Legs of effort were further subdivided into 10km-long segments to homogenize the length of sampling unit and to allow for a fine-scaled examination of differences between the two platforms. Segments were paired by temporal matching between Megafauna and Scans platforms, and the difference of encounter rates between the two platforms per category was computed for each segment.**"*

[Section 2.4.1] Here and in Figure 3, you need to clarify that the 500-m buffer around the focal sighting is omnidirectional (it's a circle), whereas the 15° interval is only perpendicular to the transect line and that "15°" refers to the angle of declination from the horizon to the sighting.

We modified the text as follows: (Lines 149-150) *"A sighting from platform 2 was tested for duplication only if it was recorded within an **omnidirectional 500 m buffer** around the focal sighting from platform 1"* and (Lines 155-157) *"If the detection angle of a candidate sighting was not included within an interval of 15° around the focal sighting detection angle (**detection angles are angles of declination recorded perpendicular to the track line**), the candidate sighting was not classified as a duplicate of the focal sighting"*.

How were declination angles measured? I am interpreting 15° to be your accepted measurement error at 600 ft altitude. Quick math tells me that the perpendicular distance between a declination of 90 deg vs. 75 deg at 600 ft alt is approximately 50 m, whereas the distance between 75 deg and 60 deg is approximately 60-ish m. I think it might help the reader understand your matching criteria if you could insert some helpful insights into the linear distances (double-check my math!) corresponding to the declination parameter. I also think it's worthwhile to somewhere state that you recognize that measurement error exists in declination angles and that is one reason for needing to implement this hierarchical method for finding duplicates.

Detection angles were measured with an inclinometer. We added details on this in the Methods section: *"(perpendicular to the track line; measured with an inclinometer)"* line 106 and *"(as for Scans platform)"* lines 115-116.

We also added a statement about the reasons behind the 15° threshold (Lines 163-165): *"The 15° criterion is necessary due to the unavoidable uncertainty in measuring declination angle, which could be due to observers spotting different animals in the sighted group and from uncertainty in assessing exactly when the sighted individuals crossed the line perpendicular to the transect"*.

[L169-172] Here you state that availability bias was assumed constant across platforms. As I noted above, this is true only if objects had the same time-in-view for both platforms. Were the windows the same size and shape? Were the fields of view identical? This needs to be stated explicitly to back up the claim/justify the assumption that the availability bias cancels out.

We added details on this matter lines 174-178: *"Availability was assumed to be the same between platforms, **justified by the bubble windows being of similar width and shape and thus providing equivalent fields of view between platforms. The time lag between platforms was very short allowing the field of view to be assumed to be the same during the time the aircraft flew over the animals.** As a consequence, the detection probability p estimated here equals the perception probability"*.

[L185] Not sure if it's worth noting, but I think the 30-sec threshold might need to be adjusted in other situations where animals might be found in very large aggregations (e.g, groups of 3000 belugas) or where clusters of 1-10 individuals might come at you rapid-fire in a foraging area. If one of your

readers is surveying in one of those cases, they shouldn't take your 30-sec threshold to be universally good!

We expanded our discussion about the 30-sec threshold to this potentially poor transferability (Lines 192-195): *"The 30 seconds threshold was chosen based on observer experience, from which the detection of a sighting was believed to have no effect on subsequent detection after this delay, and must be adjusted to particular situations when transferring the method to other study areas and species"*.

[L198-206] I'd move this entire paragraph to be the very last paragraph of your intro. This lists your hypotheses – it's helping to define the challenge you face as you work through the story that you present in the paper. This should come right after your goals ("aim") and objectives.

A lighter version of this paragraph was added to the end of the introduction (Lines 59-64).

[L226] what model selection criteria did you use to select the half-normal? I think that should be stated in methods. I think you should move the statement about selecting the half-normal to the results section

AIC was used to choose between the two models. We modified the text as follows (Lines 235-237): *"We tested half-normal and hazard rate keys to fit detection functions (Buckland et al. 2015), and selected best models using Akaike Information Criterion"* and (Line 330): *"The half-normal key was selected as the best fit for detection functions of all taxonomic groups"*.

[Section 2.5.2] Encounter rates of cetaceans could be correlated with "class" if there are mixed species hotspots or observation effects (e.g., sea state, low clouds, fog) that affect perception bias.

We did not expect a correlation between encounter rates and sighting conditions, because we limited our analysis to data from the best observation conditions. However, we expected encounter rates of cetaceans to be negatively correlated with "class" of seabirds and anthropogenic objects, as can be seen in Figure 8, due to their opposite spatial patterns in distribution (cetaceans offshore, seabirds and anthropogenic objects mostly inshore). We believe that this correlation did not impair our results, because we were not interested in the absolute abundance estimate deriving from each class, but in the difference in estimates between the two protocols.

Another question for this section: what was the sample unit for estimating encounter rate and its associated variance in this analysis?

The same procedure as in 2.5.1 was used over subsets of effort for each class, *i.e.* encounter rates, density estimates and ESWs were estimated using legs of homogeneous detection conditions as sampling unit. As above, Fewster's R2 variance estimator was used.

If you based it on something smaller than a single transect line, what did you do to investigate and address, if necessary, spatial autocorrelation?

Spatial autocorrelation is ubiquitous in ecological data, but accounting for it also depends on study objectives. Ignoring spatial autocorrelation in ecological data when it is present will result in incorrect inference of statistical significance (Dormann 2007). But this is not our aim here and bearing in mind that spatial autocorrelation is a statistical pattern that can arise from many ecological processes of interest, we think it is not necessary that it is accounted for in our analysis. We are interested in

estimating from the data, areas with contrasted encounter rates of different taxa/groups to stratify in a post-hoc manner the response variable of interest, namely detection of cetaceans. The primary aim here was to investigate whether any difference between the two protocols may be correlated with encounter rates of other taxa/groups.

[L250] I don't understand why you needed to segment the data for the hierarchical model. If I understand your model correctly, it takes point data as the input.

We thank the reviewer for pointing out this typo. This was corrected in the revised manuscript (Line 260-261): "*A hierarchical model with cross-classified random effects of observers and species for each platform was built assuming a half-normal key for the detection function*"

[L252-253] What is meant by "shrunk towards a common detection function"?

Random effects are in effect shrinkage estimators: the estimate is biased toward the mean of the assumed model. It is in that sense that detection functions were shrunk toward a common mean. The classical outlook is that of regularization, and it provides a bridge between Bayesians and frequentists (see for example Gelman and Shalizi 2013, page 19; or Gelman et al. 2014 pages 93-94). We added some explanations in the text (Lines 261-267): "*Random effects allow the leverage of partial pooling (a.k.a. borrowing strength) by shrinking each combination of platform-observer-species detection function towards a common mean detection (Virgili et al., 2019). If, for a given combination of parameters, there are few sightings, the estimated detection function will be very close to the common detection function, whereas if there are enough data, the estimated detection function can deviate from this common function. In other words, random effects allowed us to stabilise the estimated detection function even in cases of sparse data for some combinations of platform-observer-species.*"

[Section 2.5.3] I'm having trouble understanding your model structure here. I understand that you're using Bayesian methods, so I expect to see definitions for a likelihood function and prior distribution(s), along with possibly some hyperpriors. Is a "semi-normal" distribution the same thing as a "half-normal" distribution?

Thanks for pointing out this confusion. A semi-normal and half-normal distribution are in fact the same. We changed semi-normal to half normal for clarity.

Your data are the perpendicular distances, so those belong in the detection function, which I think you've defined to be "semi-normal." How did you define the mean and variance of the semi-normal? I think the variance is defined by your σ_{jkl} .

σ_{jkl} is the square-root of the variance of the assumed half-normal distribution (dispersion parameter). The mean is usually assumed to be 0 so that detection on the transect line is 1. We have clarified this point in the revised version of the manuscript Lines 275-278: "*Normal distributions were parametrised in terms of location (mean μ) and scale (standard deviation σ) parameters. We used half Student-t distributions with three degrees of freedom and scale set to 1.5 as priors for the dispersion parameters, and standard normal priors for all other parameters (see code in Supplementary File A)*"

If so, that means that your delta, eta, and gamma each need a prior distribution and probably some hyperpriors. Why did you model the random effects using a bivariate distribution? Are you assuming that observer and species effects might be correlated? I got really, really confused here.

We used bivariate normal distributions for observer and species random effects because we allowed for a possible correlation across the two platforms: the two observer teams were flying in the same plane, and species were detected from the same plane. We have clarified this point in the manuscript (Lines 272-278): " *η and γ were modelled with bivariate Normal random effects, specified with a Cholesky decomposition and using priors for the Cholesky factors from Kinney and Dunson (2008). Using a bivariate makes allowance for a correlation between platforms as the same observers rotated across platforms, and the same species could be detected from each platform. Normal distributions were parametrised in terms of location (mean μ) and scale (standard deviation σ) parameters. We used half Student-*t* distributions with three degrees of freedom and scale set to 1.5 as priors for the dispersion parameters, and standard normal priors for all other parameters (see code in Supplementary File A).*".

Lastly, why did you choose to use Bayesian inference here? My understanding is that there wasn't a considerable amount of prior information that could be used to inform the priors. Why go to the hassle of a Bayesian analysis if the rest of the paper was frequentist and you don't have a pool of relevant data for developing priors?

We used Bayesian methods for the comparison of detection functions between the two platforms because it was straightforward to fit the model we were interested in in this framework (it was not a hassle), and because it provided regularised estimates (see answer above). A recent example of this pragmatism is Boyd et al. (2019). We have used weakly-informative priors throughout.

[L272] I didn't understand the following sentence: "Marine mammals and other marine fauna had a null mean difference, while mean difference for anthropogenic objects was slightly greater for the Scans platform."

We agree this sentence was unclear and modified it (Lines 294-297): "*The mean difference in encounter rates between platforms was null for marine mammals and other marine fauna (Figure 5B). The mean difference was slightly negative for anthropogenic objects (i.e., higher encounter rate for the Scans platform), but was not statistically different from zero (Figure 5B)*".

[L309] This states, "CDS-based ESW estimation indicated a platform-independent ESW for all taxonomic groups...." However, the fact that the confidence intervals overlap almost entirely in Figure 7 doesn't seem to support this statement.

Again, the sentence was not clear so we edited it as follows (Line 330-332): "*CDS-based ESW estimation indicated there was no platform effect on ESW for any of the taxonomic groups (overlapping confidence intervals, ...)*".

[L355-359] I got a little confused by this paragraph. I think you're just trying to say that your perception probabilities were consistent with those estimated for other surveys. I would trim this down to a single sentence and move it right before the sentence beginning "Our results highlighted a slightly higher..." that is currently on L345.

The first sentence of the paragraph was moved where suggested and modified (Lines 366-369): *“According to the literature, small cetacean perception probability is survey-dependent, ranging from 47% to 96% (Carretta et al., 1998; Pollock et al., 2006; Fuentes et al., 2015). The perception probabilities estimated for the SCANS-III survey were consistently within this range, and varied between the two platforms. Our results highlighted a slightly higher perception probability...”*.

The second sentence was introducing the next paragraph so was moved there and rephrased (Lines 380-382): *“Perception of small cetaceans has been demonstrated to be affected by environmental observation conditions, and to be different among species (Marsh and Sinclair, 1989; Carretta et al., 1998; Slooten et al., 2004). Here, we assessed...”*.

[L364-368] Here you're hypothesizing why cetacean presence in the previous 30 sec induced a higher perception probability for both platforms. You suggest that the result is consistent with what could result from spatially heterogeneous cetacean density in your study area. However, I don't think that high density patches alone can explain the higher *perception* probability. Areas of high density could very likely explain the probability that another cetacean is encountered in the next 30 sec, but that is different than perception probability (i.e., $p(\text{detect}|\text{encounter})$).

We completely agree with you about this point, the higher concentration of cetaceans increases the detection probability, not the perception. However, what we were trying to argue here is that higher concentration of cetaceans implies a stronger focusing of observers on the sea surface and sub-surface, thereby increasing the perception. We modified our sentence to clarify (Lines 384-390): *“This effect was clearly the strongest identified effect and was robust to variations in the assumptions made to identify duplicates. It could be due to a real impact of cetaceans on the detection process by focusing the observer on the sea surface. **This impact of cetaceans on the detection process might be driven by a non-random distribution** of small cetaceans, known to exhibit aggregative behaviours, whereby the detection of one small cetacean group will tend to increase the probability of detecting another one **in a short period of time, thereby keeping the observer focused on the sea surface.** How influential this is would depend on the spatial scale of aggregation of these species, about which little is known.”*

[L369] We use seabirds as a cue all the time for marine mammal aerial surveys conducted in the Arctic. Sometimes gray whales that feed benthically stir up prey that seabirds can reach from the surface. Sometimes the cetaceans and seabirds are all feeding on a near-surface bait ball. I think the effect of allowing an observer's gaze to stop momentarily on a seabird could easily increase or decrease perception probability.

Multiple-species aggregations are not commonly observed in our study area, but we agree that the orientation of seabird effect on cetacean perception completely depends on studied systems, as argued further down about North Sea and tropical waters (Lines 443-458).

In addition, we now clarify that our statements about the hypothesised negative effect of seabirds applies to our study area and European shelf seas in general (Lines 391-394): *“**In European shelf seas, it has been suggested that the presence of birds could divert the attention of observers away from cetaceans or even be present in densities such that they could partially obscure the sea surface and lower the perception of cetaceans, so we expected a negative effect of seabird detection on both platforms in our study area.**”*

[L400-408] This section confused me a little. My biggest concern was that the CDS density estimator says pretty clearly that, all other things being equal, a decrease in perception probability should result in a higher density estimate. Therefore, the fact that your density estimates were comparable even though you detected slight differences in perception probabilities leads me to believe that there must have been platform-specific differences in encounter rate or group size estimates. Based on the math, something needs to compensate for the differences in p.

In fact, we believe that the effect of the perception probability might be negligible compared to the effect of distance to the transect line (as detailed in the next paragraph), and is probably hidden within the confidence intervals around the estimates. We modified the text accordingly (Lines 428-431): ***“This indicates that the variation in overall perception probabilities is probably negligible compared to the uncertainty surrounding the estimates (confidence intervals). Therefore, the differences in perception probabilities between platforms does not impact the resulting population density estimates as obtained through CDS”***

[L409-415] I found this paragraph to be very repetitive of information said multiple times previously. I'd delete it.

We have deleted the first sentences, but we moved the end of the paragraph to the end to the previous sentence (Lines 430-432): ***“Therefore, the differences in perception probabilities between platforms do not impact the resulting population density estimates as obtained through CDS, and we are confident that the negative impact of seabird detections does not affect small cetacean population assessment derived from the Megafauna protocol”***. The second sentence was retained in the fully developed paragraph (see next comment).

[L415-418] I agree with your concern here that an mcds analysis would be better. This gets back to my comment about section 2.5.2 (item #56 in the list above). Could you implement mcds just for something like sea state? I find it really hard to believe that sea states in the range of Beaufort 0 to 4 have no effect on detectability of small cetaceans, even at 600 ft survey altitude! You could simplify things by pooling Beaufort 0-2 and 3-4, thereby creating only 2 categories.

I apologise sincerely for this stupid mistake, and thank both reviewers for their kindness on this matter: I did not submit the very latest version of the paper (where this paragraph is fully developed), but the previous version, where it was not. With this section fully developed (in bold) and the previous modifications made, the paragraph now stands as follows (Lines 433-442):

“The absence of bias in population estimates through CDS might further indicate that this effect could be negligible on cetacean detection in our study area compared to other parameters accounted for in distance sampling analyses, such as distance from the transect line which is recognised as the main parameter affecting animal detection (Buckland et al., 2015), or environmental observations conditions such as Beaufort sea-state (e.g. Marsh and Sinclair, 1989; Pollock et al., 2006). A more precise abundance estimation could have been obtained by considering observation conditions, such as Beaufort sea state or subjective conditions, in a multi-covariate distance sampling (MCDS), but such analysis was precluded here by the reduced sampled size for most studied species. It was tested on common/striped dolphin groups (the only species group with sufficient sample size), and resulted in similar density estimates as from using CDS, so we are confident that using MCDS instead of CDS would not impact the conclusions presented here.”

[L442-448] I disagree with the statement that you'd need to have a given observer operating each protocol over the exact same configuration of sm cetacean sightings in order to investigate perception probability at the observer level. If there is a non-trivial effect, I think you'd just need a good sample size and you could evaluate this statistically. Or create a video game and see how distracted they get when you ask them to do multi-species tasks as opposed to focusing on a single species!

Following comments of both reviewers about this paragraph, we deleted it.

Figure 1: Insert a scale bar.

The scale bar was inserted as requested.

Figure 3b: The purpose of this figure is to help the reader visualize the matching algorithm. The matching algorithm addresses a 3-Dimensional problem (time plus space in 2D), but you can get away with illustrating just the 2D spatial components. This figure seems to plot all sightings in 1D – they are all plotted at the point on the transect line perpendicular to the actual sighting. I think this figure would be much more intuitive if you plotted the sightings perpendicular to the transect at the location specified by their respective declination angles.

We chose to plot the positions of sighting records (on the track line), instead of “true” positions, because this is the information we actually used in the algorithm. As explained in the method, we did not reconstruct positions based on declination angle and aircraft position due to GPS errors. In that way, the Figure illustrates the fact that the 500 m buffer is here to incorporate both the lag in sighting recording between platforms (recording position not on the exact same place along the transect), but also the GPS intervals that might occur. In fact, the latter is the most important reason behind the use of an omnidirectional buffer, because we sometimes had up to nearly 400 m between the two transect lines due to GPS measurement errors.

Figures 5-9: Need to clarify what the error bars represent. 95% CIs?

Thank you for having spotted this oversight. Yes, bars are 95% CI in Fig 6-9, but standard deviation in Fig5. We added this to the captions.

Reviewer 2

1) When estimating the perception probability why was distance from the track line not used? It seems a potentially very influential factor that is accounted for when using CDS, so why not in the perception probability estimation?

Distance from the track line is probably influencing perception probability, and we might expect the difference in perception between platform to change with increasing distance. Similarly, the factors often used in MCDS, such as sea-state, would probably have an effect on perception. However, to test this aspect would necessitate a larger sample size than the one collected here, so we were not able to address it.

2) Line 156: it would be good to explain that 90 degrees is straight down not out to the horizon.

We modified the text as follows to incorporate the suggestion (Lines 158-161): *"If the detection angle was greater than 85° (**sightings under the plane**), the candidate sighting was considered a duplicate. If the detection angle was less than 85° (**farther from the line, up to the horizon**), the candidate sighting was considered a duplicate if it was recorded on the same side of the aircraft as the focal sighting."*

3) Line 260: did you say what x is in the ESW equation? And what is σ ?

x is the perpendicular distance that is integrated over the range 0-truncation distance to estimate the ESW. We added it in the text Line 282-283: *"ESW for each combination of platform-observer-species was computed as (**x is the perpendicular distance**, w_k is truncation distance for species k)"*). σ is the dispersion parameter of the half-normal distribution. We again added detail in the text Line 275-276: *"Normal distributions were parametrised in terms of location (mean μ) and scale (standard deviation σ)"*.

4) Lines 378: this counterbalance seems important and influencing the overall conclusions. Seems accounting for the glare difference between the two platforms would be important and potentially shift your conclusions.

Yes, we agree. We were not able to test for this aspect because the two platforms never shifted positions inside the plane, but an interesting perspective would be to re-do the experiment with inverted position of platforms inside the plane. We added a few words about this topic Lines 377-379: *"Operating a similar double-platform experiment as in the present study but shifting the position of the two platforms inside the plane would allow us to test for the effect of direct glare on perception."*

5) Lines 416-418: yes these would be very good statements to include. It seems you do have a large enough sample to do a MCDS analysis. But maybe not? Also why did you not try out a MRDS (mark recapture distance sampling) analysis with the two teams within the area and species overlapping between the two teams, even though they used two different procedures? This of course would not have addressed your primary questions, but would you of resulted in a better final abundance estimate.

As in responding to reviewer 1's comment, I apologise sincerely for the mistake of having not submitted the very latest version of the paper (where this paragraph was fully developed), but the previous version, where it was not. The fully developed paragraph should hopefully answer the concern about MCDS (Lines 433-442). We thank the reviewer for the suggestion of using MRDS instead. We agree that fitting a MRDS model would be interesting in its own right, but, as stated, it is in fact a different question. The data we used is from an experiment that is unlikely to be repeated, and the primary question was about possible bias of the Megafauna protocol. To answer that question we chose to analyse the data as if only one of the two protocol had been implemented, which is more realistic as it is unlikely that double platform experiment with differing protocols will happen again. Hence our choice not to carry out a MRDS analysis, which we agree would yield a better estimate but would not answer our question of interest.

6) Lines 419+: It seems the fact that seabirds and cetaceans showed relatively little spatial overlap in their distributions has a huge effect on the conclusions about seabirds. Perhaps to look at the effects of

seabirds you have to only use the portions of the track lines where there was overlap. Given the reported impression in areas in high bird density areas in lines 423-426, you need to be much more cautionary about any conclusions relative to sea birds.

This discrepancy between seabird and cetacean distributions was the motivation for performing the post-stratification analysis, which allowed us to test for the persistence of the similarity in abundance estimate even in areas of overlapping density of seabirds and cetaceans (and other targets). Our results show that the density estimate for cetaceans were still very similar even when seabirds were in high concentrations. We are aware that our seabird and cetacean density tested through this post-stratification are not similar to the very high ones observed in systems like the North Sea, hence the cautions in the paragraphs Lines 443-458.

7) Lines 442-448: I'm not sure this paragraph is needed or else I missed the major point you wanted to make in it.

Following comments of both reviewers about this paragraph, we deleted it.

References:

- Boyd, C.; Hobbs, R. C.; Punt, A. E.; Shelden, K. E.; Sims, C. L. & Wade, P. R. (2019) Bayesian Estimation of Group Sizes for a Coastal Cetacean Using Aerial Survey Data. *Marine Mammal Science*
- Buckland, S. T.; Anderson, D. R.; Burnham, K. P.; Laake, J. L.; Borchers, D. L. & Thomas, L. (2001) *Introduction to Distance Sampling: Estimating Abundance of Biological Populations*. Oxford University Press
- Dormann, C. (2007) Effects of Incorporating Spatial Autocorrelation into the Analysis of Species Distribution Data. *Global Ecology and Biogeography*, 16, 129-138
- Gelman, A.; Carlin, J. B.; Stern, H. S.; Dunson, D. B.; Vehtari, A. & Rubin, D. B. (2014) *Bayesian Data Analysis* CRC Press.
- Gelman, A. & Shalizi, C. (2013) Philosophy and the Practice of Bayesian Statistics. *British Journal of Mathematical and Statistical Psychology*, 66, 8-38